# Neural Contextual Bandits with Deep Representation and Shallow Exploration

**Pan Xu**
California Institute of Technology
panxu@caltech.edu

**Zheng Wen**
DeepMind
zhengwen@google.com

**Handong Zhao**
Adobe Research
hazhao@adobe.com

**Quanquan Gu**
University of California, Los Angeles
qgu@cs.ucla.edu

## Abstract

We study neural contextual bandits, a general class of contextual bandits, where each context-action pair is associated with a raw feature vector, but the specific reward generating function is unknown. We propose a novel learning algorithm that transforms the raw feature vector using the last hidden layer of a deep ReLU neural network (deep representation learning), and uses an upper confidence bound (UCB) approach to explore in the last linear layer (shallow exploration). We prove that under standard assumptions, our proposed algorithm achieves $\widetilde{O}(\sqrt{T})$ finite-time regret, where $T$ is the learning time horizon. Compared with existing neural contextual bandit algorithms, our approach is computationally much more efficient since it only needs to explore in the last layer of the deep neural network.

## 1 Introduction

Multi-armed bandits (MAB) (Auer et al., 2002; Audibert et al., 2009; Lattimore & Szepesvári, 2020) are a class of online decision-making problems where an agent needs to learn to maximize its expected cumulative reward by repeatedly interacting with a partially known environment. Following a bandit algorithm (also called a strategy or policy), in each round, the agent adaptively chooses an arm, and then receives a reward associated with that arm. Since only the reward of the chosen arm will be observed (bandit information feedback), a good bandit algorithm has to deal with the exploration-exploitation dilemma: trade-off between pulling the best arm based on existing knowledge/history data (exploitation) and trying the arms that have not been fully explored (exploration).

In many real-world applications, the agent will also be able to access detailed contexts associated with the arms. For example, when a company wants to choose an advertisement to present to a user, the recommendation will be much more accurate if the company takes into consideration the contents, specifications, and other features of the advertisements in the arm set as well as the profile of the user. To encode the contextual information, contextual bandit models and algorithms have been developed, and widely studied both in theory and in practice (Dani et al., 2008; Rusmevichientong & Tsitsiklis, 2010; Li et al., 2010; Chu et al., 2011; Abbasi-Yadkori et al., 2011). Most existing contextual bandit algorithms assume that the expected reward of an arm at a context is a linear function in a known context-action feature vector, which leads to many useful algorithms such as LinUCB (Chu et al., 2011), OFUL (Abbasi-Yadkori et al., 2011), etc. The representation power of the linear model can be limited in applications such as marketing, social networking, clinical studies, etc., where the rewards are usually counts or binary variables. The linear contextual bandit problem has also been extended to richer classes of parametric bandits such as the generalized linear bandits (Filippi et al., 2010; Li et al., 2017) and kernelised bandits (Valko et al., 2013; Chowdhury & Gopalan, 2017).

With the prevalence of deep neural networks (DNNs) and their phenomenal performances in many machine learning tasks (LeCun et al., 2015; Goodfellow et al., 2016), there has emerged a line of work that employs DNNs to increase the representation power of contextual bandit algorithms (Allesiardo et al., 2014; Riquelme et al., 2018; Collier & Llorens, 2018; Zahavy & Mannor, 2019;

Zhou et al., 2020; Deshmukh et al., 2020; Zhang et al., 2020). The problems they solve are usually referred to as *neural contextual bandits*. For example, Zhou et al. (2020) developed the NeuralUCB algorithm, which can be viewed as a natural extension of LinUCB (Chu et al., 2011; Abbasi-Yadkori et al., 2011), where they use the output of a deep neural network with the feature vector as input to approximate the reward. Zhang et al. (2020) adapted neural networks in Thompson Sampling (Thompson, 1933; Chapelle & Li, 2011; Russo et al., 2018) for both exploration and exploitation and proposed NeuralTS . For a fixed time horizon $T$, it has been proved that both NeuralUCB and NeuralTS achieve a $O(\widetilde{d}\sqrt{T})$ regret bound, where $\widetilde{d}$ is the effective dimension of a neural tangent kernel matrix which can potentially scale with $O(TK)$ for $K$-armed bandits. This high complexity is mainly due to that the exploration is performed over the entire huge neural network parameter space, which is inefficient and even infeasible when the number of neurons is large. A more realistic and efficient way of learning neural contextual bandits may be to just explore different arms using the last layer as the exploration parameter. More specifically, Riquelme et al. (2018) provided an extensive empirical study of benchmark algorithms for contextual-bandits through the lens of Thompson Sampling, which suggests decoupling representation learning and uncertainty estimation improves performance.

In this paper, we show that the decoupling of representation learning and the exploration can be theoretically validated. We study a new neural contextual bandit algorithm, which learns a mapping to transform the raw features associated with each context-action pair using a deep neural network (*deep representation*), and then performs an upper confidence bound (UCB)-type exploration over the linear output layer of the network (*shallow exploration*). We prove a sublinear regret of the proposed algorithm by exploiting the UCB exploration techniques in linear contextual bandits (Abbasi-Yadkori et al., 2011) and the analysis of deep overparameterized neural networks using neural tangent kernels (Jacot et al., 2018). Our theory confirms the empirically observed effectiveness of decoupling the deep representation learning and the UCB exploration in contextual bandits (Riquelme et al., 2018; Zahavy & Mannor, 2019).

**Contributions** we summarize the main contributions of this paper as follows.

- We propose a contextual bandit algorithm, Neural-LinUCB, for solving a general class of contextual bandit problems without knowing the specific reward generating function. The proposed algorithm learns a deep representation to transform the raw feature vectors and performs UCB-type exploration in the last layer of the neural network, which we refer to as deep representation and shallow exploration. Compared with LinUCB (Li et al., 2010; Chu et al., 2011) and neural bandits such as NeuralUCB (Zhou et al., 2020) and NeuralTS (Zhang et al., 2020), our algorithm enjoys the best of two worlds: strong expressiveness due to the deep representation and computational efficiency due to the shallow exploration.
- Despite the usage of a DNN as the feature mapping, we prove a $\widetilde{O}(\sqrt{T})$ regret for the proposed Neural-LinUCB algorithm, which matches the regret bound of linear contextual bandits (Chu et al., 2011; Abbasi-Yadkori et al., 2011). To the best of our knowledge, this is the first work that theoretically shows the convergence of bandits algorithms under the scheme of deep representation and shallow exploration. It is notable that a similar scheme called Neural-Linear was proposed by Riquelme et al. (2018) for Thompson sampling algorithms, and they empirically showed that decoupling representation learning and uncertainty estimation improves the performance. Our work confirms this observation from a theoretical perspective.
- We conduct experiments on contextual bandit problems based on real-world datasets, demonstrating a better performance and computational efficiency of Neural-LinUCB over LinUCB and existing neural bandits algorithms such as NeuralUCB, which well aligns with our theory.

## 1.1 ADDITIONAL RELATED WORK

There is a line of related work to ours on the recent advance in the optimization and generalization analysis of deep neural networks. In particular, Jacot et al. (2018) first introduced the neural tangent kernel (NTK) to characterize the training dynamics of network outputs in the infinite width limit. From the notion of NTK, a fruitful line of research emerged and showed that loss functions of deep neural networks trained by (stochastic) gradient descent can converge to the global minimum (Du et al., 2019b; Allen-Zhu et al., 2019b; Du et al., 2019a; Zou et al., 2018; Zou & Gu, 2019). The generalization bounds for overparameterized deep neural networks are also established in Arora et al. (2019a;b); Allen-Zhu et al. (2019a); Cao & Gu (2019a;b). Recently, the NTK based analysis

is also extended to the study of sequential decision problems including bandits (Zhou et al., 2020; Zhang et al., 2020), and reinforcement learning algorithms (Cai et al., 2019; Liu et al., 2019; Wang et al., 2020; Xu & Gu, 2020).

Our algorithm is also different from Langford & Zhang (2008); Agarwal et al. (2014) which reduce the bandit problem to supervised learning. Moreover, their algorithms need to access an oracle that returns the optimal policy in a policy class given a sequence of context and reward vectors, whose regret depends on the VC-dimension of the policy class.

**Notation** We use $[k]$ to denote a set $\{1, \ldots, k\}$, $k \in \mathbb{N}^+$. $\|\mathbf{x}\|_2 = \sqrt{\mathbf{x}^\top \mathbf{x}}$ is the Euclidean norm of a vector $\mathbf{x} \in \mathbb{R}^d$. For a matrix $\mathbf{W} \in \mathbb{R}^{m \times n}$, we denote by $\|\mathbf{W}\|_2$ and $\|\mathbf{W}\|_F$ its operator norm and Frobenius norm respectively. For a semi-definite matrix $\mathbf{A} \in \mathbb{R}^{d \times d}$ and a vector $\mathbf{x} \in \mathbb{R}^d$, we denote the Mahalanobis norm as $\|\mathbf{x}\|_\mathbf{A} = \sqrt{\mathbf{x}^\top \mathbf{A} \mathbf{x}}$. Throughout this paper, we reserve the notations $\{C_i\}_{i=0,1,\ldots}$ to represent absolute positive constants that are independent of problem parameters such as dimension, sample size, iteration number, step size, network length and so on. The specific values of $\{C_i\}_{i=0,1,\ldots}$ can be different in different context. For a parameter of interest $T$ and a function $f(T)$, we use notations such as $O(f(T))$ and $\Omega(f(T))$ to hide constant factors and $\widetilde{O}(f(T))$ to hide constant and logarithmic dependence of $T$.

## 2 PRELIMINARIES

In this section, we provide the background of contextual bandits and deep neural networks.

### 2.1 LINEAR CONTEXTUAL BANDITS

A contextual bandit is characterized by a tuple $(\mathcal{S}, \mathcal{A}, r)$, where $\mathcal{S}$ is the context (state) space, $\mathcal{A}$ is the arm (action) space, and $r$ encodes the unknown *reward generating function* at all context-arm pairs. A learning agent, who knows $\mathcal{S}$ and $\mathcal{A}$ but does not know the true reward $r$ (values bounded in $(0, 1)$ for simplicity), needs to interact with the contextual bandit for $T$ rounds. At each round $t = 1, \ldots, T$, the agent first observes a context $s_t \in \mathcal{S}$ chosen by the environment; then it needs to adaptively select an arm $a_t \in \mathcal{A}$ based on its past observations; finally it receives a reward

$$\widehat{r}_t(\mathbf{x}_{s,a_t}) = r(\mathbf{x}_{s,a_t}) + \xi_t, \tag{2.1}$$

where $\mathbf{x}_{s,a} \in \mathbb{R}^d$ is a known feature vector for context-arm pair $(s, a) \in \mathcal{S} \times \mathcal{A}$, and $\xi_t$ is a random noise with zero mean. The agent's objective is to maximize its expected total reward over these $T$ rounds, which is equivalent to minimizing the pseudo regret (Audibert et al., 2009):

$$R_T = \mathbb{E}\left[\sum_{t=1}^{T} \left(\widehat{r}(\mathbf{x}_{s_t,a_t^*}) - \widehat{r}(\mathbf{x}_{s_t,a_t})\right)\right], \tag{2.2}$$

where $a_t^* \in \operatorname{argmax}_{a \in \mathcal{A}}\{r(\mathbf{x}_{s_t,a}) = \mathbb{E}[\widehat{r}(\mathbf{x}_{s_t,a})]\}$. To simplify the exposition, we use $\mathbf{x}_{t,a}$ to denote $\mathbf{x}_{s_t,a}$ since it only depends on the round index $t$ in most bandit problems, and we assume $\mathcal{A} = [K]$.

In linear contextual bandits, the reward function in (2.1) is assumed to have a linear structure $r(\mathbf{x}_{s,a}) = \mathbf{x}_{s,a}^\top \boldsymbol{\theta}^*$ for some unknown weight vector $\boldsymbol{\theta}^* \in \mathbb{R}^d$. One provably sample efficient algorithm for linear contextual bandits is Linear Upper Confidence Bound (LinUCB) (Chu et al., 2011) or Optimism in the Face of Uncertainty Linear bandit algorithm (OFUL) (Abbasi-Yadkori et al., 2011). Specifically, at each round $t$, LinUCB chooses the action $a_t = \operatorname{argmax}_{a \in [K]}\{\mathbf{x}_{t,a}^\top \boldsymbol{\theta}_t + \alpha_t \|\mathbf{x}_{t,a}\|_{\mathbf{A}_t^{-1}}\}$, where $\boldsymbol{\theta}_t$ is a point estimate of $\boldsymbol{\theta}^*$, $\mathbf{A}_t = \lambda \mathbf{I} + \sum_{i=1}^{t} \mathbf{x}_{i,a_i} \mathbf{x}_{i,a_i}^\top$ with some $\lambda > 0$ is a matrix defined based on the historical context-arm pairs, and $\alpha_t > 0$ is a tuning parameter that controls the exploration rate in LinUCB.

### 2.2 DEEP NEURAL NETWORKS

In this paper, we use $f(\mathbf{x})$ to denote a neural network with input data $\mathbf{x} \in \mathbb{R}^d$. Let $L$ be the number of hidden layers and $\mathbf{W}_l \in \mathbb{R}^{m_l \times m_{l-1}}$ be the weight matrices in the $l$-th layer, where $l = 1, \ldots, L$, $m_1 = \ldots = m_{L-1} = m$ and $m_0 = m_L = d$. Then a $L$-hidden layer neural network is defined as

$$f(\mathbf{x}) = \sqrt{m} \boldsymbol{\theta}^{*\top} \sigma_L(\mathbf{W}_L \sigma_{L-1}(\mathbf{W}_{L-1} \cdots \sigma_1(\mathbf{W}_1 \mathbf{x}) \cdots)), \tag{2.3}$$

where $\sigma_l$ is an activation function and $\boldsymbol{\theta}^* \in \mathbb{R}^d$ is the weight of the output layer. To simplify the presentation, we will assume $\sigma_1 = \sigma_2 = \ldots = \sigma_L = \sigma$ is the ReLU activation function, i.e., $\sigma(x) = \max\{0, x\}$ for $x \in \mathbb{R}$. We denote $\mathbf{w} = (\text{vec}(\mathbf{W}_1)^\top, \ldots, \text{vec}(\mathbf{W}_L)^\top)^\top$, which is the concatenation of the vectorized weight parameters of all hidden layers of the neural network. We also write $f(\mathbf{x}; \boldsymbol{\theta}^*, \mathbf{w}) = f(\mathbf{x})$ in order to explicitly specify the weight parameters of neural network $f$. It is easy to show that the dimension $p$ of vector $\mathbf{w}$ satisfies $p = (L-2)m^2 + 2md$. To simplify the notation, we define $\boldsymbol{\phi}(\mathbf{x}; \mathbf{w})$ as the output of the $L$-th hidden layer of neural network $f$.

$$\boldsymbol{\phi}(\mathbf{x}; \mathbf{w}) = \sqrt{m}\sigma(\mathbf{W}_L\sigma(\mathbf{W}_{L-1}\cdots\sigma(\mathbf{W}_1\mathbf{x})\cdots)). \tag{2.4}$$

Note that $\boldsymbol{\phi}(\mathbf{x}; \mathbf{w})$ itself can also be viewed as a neural network with vector-valued outputs.

## 3 DEEP REPRESENTATION AND SHALLOW EXPLORATION

### 3.1 HIGH-LEVEL IDEA OF THE PROPOSED ALGORITHM

The linear parametric form in linear contextual bandits might produce biased estimates of the reward due to the lack of representation power (Snoek et al., 2015; Riquelme et al., 2018). In contrast, it is well known that deep neural networks are powerful enough to approximate an arbitrary function (Cybenko, 1989). Therefore, a natural extension of linear contextual bandits is to use a deep neural network to approximate the reward generating function $r(\cdot)$. Nonetheless, DNNs usually have a prohibitively large dimension for weight parameters, which makes the exploration in neural networks based UCB algorithm inefficient (Kveton et al., 2020; Zhou et al., 2020).

In this work, we study a neural contextual bandit algorithm, where the hidden layers of a deep neural network are used to represent the features and the exploration is only performed in the last layer of the neural network. In particular, for any arm feature vector $\mathbf{x}$, we use $\langle \boldsymbol{\theta}^*, \boldsymbol{\phi}(\mathbf{x}; \mathbf{w}) \rangle$ to approximate the unknown reward function $r(\mathbf{x})$, where $\boldsymbol{\phi}(\mathbf{x}; \mathbf{w})$ defined as in (2.4) is a neural network with weight $\mathbf{w}$, and $\boldsymbol{\theta}^*$ is a unknown weight parameter. Note that we can also view $\langle \boldsymbol{\theta}^*, \boldsymbol{\phi}(\mathbf{x}; \mathbf{w}) \rangle$ as a neural network with $\boldsymbol{\phi}(\mathbf{x}; \mathbf{w})$ being the output of the last hidden layer and $\boldsymbol{\theta}^*$ the weight parameter of the last (linear) layer. Different from existing neural bandit algorithms, we only add a UCB bonus term involving the last layer instead of all the weight parameter of this large neural network.

This decoupling of the representation and the exploration achieves the best of both worlds: efficient exploration of shallow (linear) models and high expressive power of deep models. In what follows, we will describe a neural contextual bandit algorithm that uses the output of the last hidden layer of a neural network to transform the raw feature vectors (*deep representation*) and performs UCB-type exploration in the last layer of the neural network (*shallow exploration*). Since the exploration is performed only in the last linear layer, we call this procedure Neural-LinUCB, which is displayed in Algorithm 1.

### 3.2 DETAILED IMPLEMENTATION OF DEEP REPRESENTATION AND SHALLOW EXPLORATION

Now we describe the details of Algorithm 1. In round $t$, the agent receives an action set with raw features $\mathcal{X}_t = \{\mathbf{x}_{t,1}, \ldots, \mathbf{x}_{t,K}\}$. Then the agent chooses an arm $a_t$ that maximizes the following upper confidence bound:

$$a_t = \underset{k \in [K]}{\operatorname{argmax}} \left\{ \langle \boldsymbol{\phi}(\mathbf{x}_{t,k}; \mathbf{w}_{t-1}), \boldsymbol{\theta}_{t-1} \rangle + \alpha_t \|\boldsymbol{\phi}(\mathbf{x}_{t,k}; \mathbf{w}_{t-1})\|_{\mathbf{A}_{t-1}^{-1}} \right\}, \tag{3.1}$$

where $\boldsymbol{\theta}_{t-1}$ is a point estimate of the unknown weight in the last layer, $\boldsymbol{\phi}(\mathbf{x}; \mathbf{w})$ is defined as in (2.4), $\mathbf{w}_{t-1}$ is an estimate of all the weight parameters in the hidden layers of the neural network, $\alpha_t > 0$ is the algorithmic parameter controlling the exploration, and $\mathbf{A}_t$ defined as follows.

$$\mathbf{A}_t = \lambda\mathbf{I} + \sum_{i=1}^{t} \boldsymbol{\phi}(\mathbf{x}_{i,a_i}; \mathbf{w}_{i-1})\boldsymbol{\phi}(\mathbf{x}_{i,a_i}; \mathbf{w}_{i-1})^\top, \tag{3.2}$$

and $\lambda > 0$. After pulling arm $a_t$, the agent will observe a noisy reward $\widehat{r}_t := \widehat{r}(\mathbf{x}_{t,a_t}) = r(\mathbf{x}_{t,k}) + \xi_t$, where $\xi_t$ is an independent $\nu$-subGaussian random noise for some $\nu > 0$ and $r(\cdot)$ is an unknown reward function. In this paper, we will interchangeably use notation $\widehat{r}_t$ to denote the reward received at the $t$-th step and an equivalent notation $\widehat{r}(\mathbf{x})$ to express its dependence on the feature vector $\mathbf{x}$.

Upon receiving the reward $\widehat{r}_t$, the agent updates its estimate $\boldsymbol{\theta}_t$ of the output layer weight by using the same $\ell^2$-regularized least-squares estimate in linear contextual bandits (Abbasi-Yadkori et al., 2011). In particular, we have $\boldsymbol{\theta}_t = \mathbf{A}_t^{-1}\mathbf{b}_t$, where $\mathbf{b}_t = \sum_{i=1}^{t} \widehat{r}_i \phi(\mathbf{x}_{i,a_i}; \mathbf{w}_{i-1})$.

To save the computation, the neural network $\phi(\cdot; \mathbf{w}_t)$ will be updated once every $H$ steps. Therefore, we have $\mathbf{w}_{(q-1)H+1} = \ldots = \mathbf{w}_{qH}$ for $q = 1, 2, \ldots$. We call the time steps $\{(q-1)H+1, \ldots, qH\}$ an epoch with length $H$. At time step $t = Hq$, we will retrain the neural network based on all the historical data via Algorithm 2, which minimizes the following empirical loss function:

$$\mathcal{L}_q(\mathbf{w}) = \sum_{i=1}^{qH} \left( \boldsymbol{\theta}_i^\top \phi(\mathbf{x}_{i,a_i}; \mathbf{w}) - \widehat{r}_i \right)^2. \tag{3.3}$$

In practice, one can further save computational cost by only feeding data $\{\mathbf{x}_{i,a_i}, \widehat{r}_i, \boldsymbol{\theta}_i\}_{i=(q-1)H+1}^{qH}$ from the $q$-th epoch into Algorithm 2 to update the parameter $\mathbf{w}_t$, which does not hurt the performance since the historical information has been encoded into the estimate of $\boldsymbol{\theta}_i$. In this paper, we will perform the following gradient descent step $\mathbf{w}_q^{(s)} = \mathbf{w}_q^{(s-1)} - \eta_q \nabla_{\mathbf{w}} \mathcal{L}_q(\mathbf{w}^{(s-1)})$, for $s = 1, \ldots, n$, where $\mathbf{w}_q^{(0)} = \mathbf{w}^{(0)}$ is chosen as the same random initialization point. We will discuss more about the initial point $\mathbf{w}^{(0)}$ in the next paragraph. Then Algorithm 2 outputs $\mathbf{w}_q^{(n)}$ and we set it as the updated weight parameter $\mathbf{w}_{Hq+1}$ in Algorithm 1. In the next round, the agent will receive another action set $\mathcal{X}_{t+1}$ with raw feature vectors and repeat the above steps to choose the sub-optimal arm and update estimation for contextual parameters.

**Initialization:** Recall that $\mathbf{w}$ is the collection of all hidden layer weight parameters of the neural network. We will follow the same initialization scheme as used in Zhou et al. (2020), where each entry of the weight matrices follows some Gaussian distribution. Specifically, for any $l \in \{1, \ldots, L-1\}$, we set $\mathbf{W}_l = \begin{bmatrix} \mathbf{W} & \mathbf{0} \\ \mathbf{0} & \mathbf{W} \end{bmatrix}$, where each entry of $\mathbf{W}$ follows distribution $N(0, 4/m)$ independently; for $\mathbf{W}_L$, we set it as $[\mathbf{V} \quad -\mathbf{V}]$, where each entry of $\mathbf{V}$ follows distribution $N(0, 2/m)$ independently.

**Comparison with LinUCB and NeuralUCB:** Compared with linear contextual bandits in Section 2.1, Algorithm 1 has a distinct feature that it learns a deep neural network to obtain a deep representation of the raw data vectors and then performs UCB exploration. This deep representation allows our algorithm to characterize more intrinsic and latent information about the raw data $\{\mathbf{x}_{t,k}\}_{t \in [T], k \in [K]} \subset \mathbb{R}^d$. However, the increased complexity of the feature mapping $\phi(\cdot; \mathbf{w})$ also introduces great hardness in training. For instance, a recent work by Zhou et al. (2020) also studied the neural contextual bandit problem, but different from (3.1), their algorithm (NeuralUCB) performs the UCB exploration on the entire network parameter space, which is $\mathbb{R}^{\widetilde{p}+d}$, where $\widetilde{p} = m + md + (L-1)m^2$. Note that in Zhou et al. (2020), they need to compute the inverse of a matrix $\mathbf{Z}_t \in \mathbb{R}^{(\widetilde{p}+d) \times (\widetilde{p}+d)}$, which is defined in a similar way to the matrix $\mathbf{A}_t$ in our paper except that $\mathbf{Z}_t$ is defined based on the gradient of the network instead of the output of the last hidden layer as in (3.2). In sharp contrast, $\mathbf{A}_t$ in our paper is only of size $d \times d$ and thus is much more efficient and practical in implementation, which will be seen from our experiments in later sections.

We note that there is also a similar algorithm to our Neural-LinUCB presented in Deshmukh et al. (2020), where they studied the self-supervised learning loss in contextual bandits with neural network representation for computer vision problems. However, no regret analysis has been provided. When the feature mapping $\phi(\cdot; \mathbf{w})$ is an identity function, the problem reduces to linear contextual bandits where we directly use $\mathbf{x}_t$ as the feature vector. In this case, it is easy to see that Algorithm 1 reduces to LinUCB (Chu et al., 2011) since we do not need to learn the representation parameter $\mathbf{w}$.

**Comparison with Neural-Linear:** The high-level idea of decoupling the representation and exploration in our algorithm is also similar to that of the Neural-Linear algorithm (Riquelme et al., 2018; Zahavy & Mannor, 2019), which trains a deep neural network to learn a representation of the raw feature vectors, and then uses a Bayesian linear regression to estimate the uncertainty in the bandit problem. However, these two algorithms are significantly different since Neural-Linear (Riquelme et al., 2018) is a Thompson sampling based algorithm that uses posterior sampling to estimate the weight parameter $\boldsymbol{\theta}^*$ via Bayesian linear regression, whereas Neural-LinUCB adopts upper confidence bound based techniques to estimate the weight $\boldsymbol{\theta}^*$. Nevertheless, both algorithms share the same idea of deep representation and shallow exploration, and we view our Neural-LinUCB algorithm as one instantiation of the Neural-Linear scheme.

---

**Algorithm 1** Deep Representation and Shallow Exploration (Neural-LinUCB)

---

1: **Input**: regularization parameter $\lambda > 0$, number of total steps $T$, episode length $H$, exploration parameters $\{\alpha_t > 0\}_{t \in [T]}$
2: **Initialization:** $\mathbf{A}_0 = \lambda \mathbf{I}$, $\mathbf{b}_0 = \mathbf{0}$; entries of $\boldsymbol{\theta}_0$ follow $N(0, 1/d)$, and $\mathbf{w}^{(0)}$ is initialized as described in Section 3; $q = 1$; $\mathbf{w}_0 = \mathbf{w}^{(0)}$
3: **for** $t = 1, \ldots, T$ **do**
4:     receive feature vectors $\{\mathbf{x}_{t,1}, \ldots, \mathbf{x}_{t,K}\}$
5:     choose arm $a_t = \mathrm{argmax}_{k \in [K]} \boldsymbol{\theta}_{t-1}^\top \boldsymbol{\phi}(\mathbf{x}_{t,k}; \mathbf{w}_{t-1}) + \alpha_t \|\boldsymbol{\phi}(\mathbf{x}_{t,k}; \mathbf{w}_{t-1})\|_{\mathbf{A}_{t-1}^{-1}}$, and obtain reward $\widehat{r}_t$
6:     update $\mathbf{A}_t$ and $\mathbf{b}_t$ as follows:
$$\mathbf{A}_t = \mathbf{A}_{t-1} + \boldsymbol{\phi}(\mathbf{x}_{t,a_t}; \mathbf{w}_{t-1}) \boldsymbol{\phi}(\mathbf{x}_{t,a_t}; \mathbf{w}_{t-1})^\top, \quad \mathbf{b}_t = \mathbf{b}_{t-1} + \widehat{r}_t \boldsymbol{\phi}(\mathbf{x}_{t,a_t}; \mathbf{w}_{t-1}),$$
7:     update $\boldsymbol{\theta}_t = \mathbf{A}_t^{-1} \mathbf{b}_t$
8:     **if** $\mathrm{mod}(t, H) = 0$ **then**
9:         $\mathbf{w}_t \leftarrow$ output of Algorithm 2
10:         $q = q + 1$
11:     **else**
12:         $\mathbf{w}_t = \mathbf{w}_{t-1}$
13:     **end if**
14: **end for**
15: **Output** $\mathbf{w}_T$

---

**Algorithm 2** Update Weight Parameters with Gradient Descent

---

1: **Input:** initial point $\mathbf{w}_q^{(0)} = \mathbf{w}^{(0)}$, maximum iteration number $n$, step size $\eta_q$, and loss function defined in (3.3).
2: **for** $s = 1, \ldots, n$ **do**
3:     $\mathbf{w}_q^{(s)} = \mathbf{w}_q^{(s-1)} - \eta_q \nabla_{\mathbf{w}} \mathcal{L}_q(\mathbf{w}_q^{(s-1)})$.
4: **end for**
5: **Output** $\mathbf{w}_q^{(n)}$

---

## 4 MAIN RESULTS

To analyze the regret bound of Algorithm 1, we first lay down some important assumptions on the neural contextual bandit model.

**Assumption 4.1.** For all $i \geq 1$ and $k \in [K]$, we assume that $\|\mathbf{x}_{i,k}\|_2 = 1$ and its entries satisfy $[\mathbf{x}_{i,k}]_j = [\mathbf{x}_{i,k}]_{j+d/2}$.

The assumption that $\|\mathbf{x}_{i,k}\|_2 = 1$ is not essential and is only imposed for simplicity, which is also used in Zou & Gu (2019); Zhou et al. (2020). The condition on the entries of $\mathbf{x}_{i,k}$ is also mild since otherwise we could always construct $\mathbf{x}'_{i,k} = [\mathbf{x}_{i,k}^\top, \mathbf{x}_{i,k}^\top]^\top / \sqrt{2}$ to replace it. An implication of Assumption 4.1 is that the initialization scheme in Algorithm 1 results in $\boldsymbol{\phi}(\mathbf{x}_{i,k}; \mathbf{w}^{(0)}) = \mathbf{0}$ for all $i \in [T]$ and $k \in [K]$.

We assume the following stability condition on the spectral norm of the neural network gradient:

**Assumption 4.2.** There is a constant $\ell_{\mathrm{Lip}} > 0$ such that it holds $\left\| \frac{\partial \boldsymbol{\phi}}{\partial \mathbf{w}}(\mathbf{x}; \mathbf{w}_0) - \frac{\partial \boldsymbol{\phi}}{\partial \mathbf{w}}(\mathbf{x}'; \mathbf{w}_0) \right\|_2 \leq \ell_{\mathrm{Lip}} \|\mathbf{x} - \mathbf{x}'\|_2$ for all $\mathbf{x}, \mathbf{x}' \in \{\mathbf{x}_{i,k}\}_{i \in [T], k \in [K]}$.

The inequality in Assumption 4.2 resembles the Lipschitz condition on the gradient of the neural network. However, it is essentially different from the smoothness condition since here the gradient is taken with respect to the neural network weights while the Lipschitz condition is imposed on the feature parameter $\mathbf{x}$. Similar conditions are widely made in nonconvex optimization (Wang et al., 2014; Balakrishnan et al., 2017; Xu et al., 2017), in the name of first-order stability, which is essential to derive the convergence of alternating optimization algorithms. Furthermore, Assumption 4.2 is only required on the $TK$ training data points and a specific weight parameter $\mathbf{w}_0$. Therefore, the condition will hold if the raw feature data lie in a certain subspace of $\mathbb{R}^d$. We provided some further discussions in the supplementary material about this assumption for interested readers.

In order to analyze the regret bound of Algorithm 1, we need to characterize the properties of the deep neural network in (2.3) that is used to represent the feature vectors. Following a recent line of research (Jacot et al., 2018; Cao & Gu, 2019a; Arora et al., 2019b; Zhou et al., 2020), we define the covariance between two data point $\mathbf{x}, \mathbf{y} \in \mathbb{R}^d$ as follows.

$$\widetilde{\boldsymbol{\Sigma}}^{(0)}(\mathbf{x}, \mathbf{y}) = \boldsymbol{\Sigma}^{(0)}(\mathbf{x}, \mathbf{y}) = \mathbf{x}^\top \mathbf{y},$$

$$\boldsymbol{\Lambda}^{(l)}(\mathbf{x}, \mathbf{y}) = \begin{bmatrix} \boldsymbol{\Sigma}^{l-1}(\mathbf{x}, \mathbf{x}) & \boldsymbol{\Sigma}^{l-1}(\mathbf{x}, \mathbf{y}) \\ \boldsymbol{\Sigma}^{l-1}(\mathbf{y}, \mathbf{x}) & \boldsymbol{\Sigma}^{l-1}(\mathbf{y}, \mathbf{y}) \end{bmatrix},$$

$$\boldsymbol{\Sigma}^{(l)}(\mathbf{x}, \mathbf{y}) = 2\mathbb{E}_{(u,v)\sim N(\mathbf{0}, \boldsymbol{\Lambda}^{(l-1)}(\mathbf{x}, \mathbf{y}))}[\sigma(u)\sigma(v)],$$

$$\widetilde{\boldsymbol{\Sigma}}^{(l)}(\mathbf{x}, \mathbf{y}) = 2\widetilde{\boldsymbol{\Sigma}}^{(l-1)}(\mathbf{x}, \mathbf{y})\mathbb{E}_{u,v}[\dot{\sigma}(u)\dot{\sigma}(v)] + \boldsymbol{\Sigma}^{(l)}(\mathbf{x}, \mathbf{y}), \tag{4.1}$$

where $(u, v) \sim N(\mathbf{0}, \boldsymbol{\Lambda}^{(l-1)}(\mathbf{x}, \mathbf{y}))$, and $\dot{\sigma}(\cdot)$ is the derivative of activation function. We denote the neural tangent kernel (NTK) matrix $\mathbf{H} \in \mathbb{R}^{TK \times TK}$ based on all feature vectors $\{\mathbf{x}_{t,k}\}_{t\in[T],k\in[K]}$. Renumbering $\{\mathbf{x}_{t,k}\}_{t\in[T],k\in[K]}$ as $\{\mathbf{x}_i\}_{i=1,\ldots,TK}$, then each entry $\mathbf{H}_{ij}$ is defined as

$$\mathbf{H}_{ij} = \frac{1}{2}\big(\widetilde{\boldsymbol{\Sigma}}^{(L)}(\mathbf{x}_i, \mathbf{x}_j) + \boldsymbol{\Sigma}^{(L)}(\mathbf{x}_i, \mathbf{x}_j)\big), \tag{4.2}$$

for all $i, j \in [TK]$. Based on the above definition, we impose the following assumption on $\mathbf{H}$.

**Assumption 4.3.** The neural tangent kernel defined in (4.2) is positive definite, i.e., $\lambda_{\min}(\mathbf{H}) \geq \lambda_0$ for some constant $\lambda_0 > 0$.

Assumption 4.3 essentially requires the neural tangent kernel matrix $\mathbf{H}$ to be non-singular, which is a mild condition and also imposed in other related work (Du et al., 2019a; Arora et al., 2019b; Cao & Gu, 2019a; Zhou et al., 2020). Moreover, it is shown that Assumption 4.3 can be easily derived from Assumption 4.1 for two-layer ReLU networks (Oymak & Soltanolkotabi, 2020; Zou & Gu, 2019). Therefore, Assumption 4.3 is mild or even negligible given the non-degeneration assumption on the feature vectors. Also note that matrix $\mathbf{H}$ is only defined based on layers $l = 1, \ldots, L$ of the neural network, and does not depend on the output layer $\boldsymbol{\theta}$. It is easy to extend the definition of $\mathbf{H}$ to the NTK matrix defined on all layers including the output layer $\boldsymbol{\theta}$, which would also be positive definite by Assumption 4.3 and the recursion in (4.2).

Before we present the regret analysis of the neural contextual bandit, we need to modify the regret defined in (2.2) to account for the randomness of the neural network initialization. For a fixed time horizon $T$, we define the regret of Algorithm 1 as follows.

$$R_T = \mathbb{E}\bigg[\sum_{t=1}^{T} \big(\widehat{r}(\mathbf{x}_{t,a_t^*}) - \widehat{r}(\mathbf{x}_{t,a_t})\big)|\mathbf{w}^{(0)}\bigg], \tag{4.3}$$

where the expectation is taken over the randomness of the reward noise. Note that $R_T$ defined in (4.3) is still a random variable since the initialization of Algorithm 2 is randomly generated.

Now we are going to present the regret bound of the proposed algorithm.

**Theorem 4.4.** Suppose Assumptions 4.1, 4.2 and 4.3 hold. Assume that $\|\boldsymbol{\theta}^*\|_2 \leq M$ for some positive constant $M > 0$. For any $\delta \in (0, 1)$, let us choose $\alpha_t$ in Neural-LinUCB as

$$\alpha_t = \nu\sqrt{2\big(d\log(1 + t\log(HK)/\lambda) + \log(1/\delta)\big)} + \lambda^{1/2}M.$$

We choose the step size $\eta_q$ of Algorithm 2 as $\eta_q \leq C_0\big(d^2 mnT^{5.5}L^6\log(TK/\delta)\big)^{-1}$ and the width of the neural network satisfies $m = \text{poly}(L, d, 1/\delta, H, \log(TK/\delta))$. With probability at least $1 - \delta$ over the randomness of the initialization of the neural network, it holds that

$$R_T \leq C_1\alpha_T\sqrt{Td\log\Big(1 + \frac{TG^2}{\lambda d}\Big)} + \frac{C_2\ell_{\text{Lip}}L^3 d^{5/2}T\sqrt{\log m \log(\frac{1}{\delta})\log(\frac{TK}{\delta})}\|\mathbf{r} - \widetilde{\mathbf{r}}\|_{\mathbf{H}^{-1}}}{m^{1/6}},$$

where constants $\{C_i\}_{i=0,1,2}$ are independent of the problem, $\mathbf{r} = (r(\mathbf{x}_1), r(\mathbf{x}_2), \ldots, r(\mathbf{x}_{TK}))^\top \in \mathbb{R}^{TK}$ and $\widetilde{\mathbf{r}} = (f(\mathbf{x}_1; \boldsymbol{\theta}_0, \mathbf{w}_0), \ldots, f(\mathbf{x}_{TK}; \boldsymbol{\theta}_{T-1}, \mathbf{w}_{T-1}))^\top \in \mathbb{R}^{TK}$, and $\|\mathbf{r}\|_{\mathbf{A}} = \sqrt{\mathbf{r}^\top \mathbf{A}\mathbf{r}}$.

**Remark 4.5.** Theorem 4.4 shows that the regret of Algorithm 1 can be bounded by two parts: the first part is of order $\widetilde{O}(\sqrt{T})$, which resembles the regret bound of linear contextual bandits (Abbasi-Yadkori et al., 2011); the second part is of order $\widetilde{O}(m^{-1/6}T\sqrt{(\mathbf{r}-\widetilde{\mathbf{r}})^\top \mathbf{H}^{-1}(\mathbf{r}-\widetilde{\mathbf{r}})})$, which depends on the estimation error of the neural network $f$ for the reward generating function $r$ and the neural tangent kernel $\mathbf{H}$.

It is worth noting that our theoretical analysis depends on the reward structure assumption that $r(\cdot) = \langle \boldsymbol{\theta}_*, \boldsymbol{\psi}(\cdot)\rangle$. However, the linear structure between $\boldsymbol{\theta}_*$ and $\boldsymbol{\psi}(\cdot)$ is not essential. As long as the deep representation of the feature vector and the uncertainty weight parameter can be decoupled, Algorithm 1 can be easily extended to settings with milder assumptions on the reward structure such as generalized linear models (Sarkar, 1991; Filippi et al., 2010; Li et al., 2017; Kveton et al., 2020). For more general bandit models where no assumption is imposed to the reward generating function, it is still unclear whether the decoupled deep representation and shallow exploration would work especially in cases a thorough exploration may be needed.

Based on the result in Theorem 4.4, we can easily verify the following conclusion:

**Corollary 4.6.** Under the same conditions of Theorem 4.4, if we choose a sufficiently overparameterized neural network mapping $\phi(\cdot)$ such that $m \geq T^3$, then the regret of Algorithm 1 is $R_T = \widetilde{O}(\sqrt{T}\sqrt{(\mathbf{r}-\widetilde{\mathbf{r}})^\top \mathbf{H}^{-1}(\mathbf{r}-\widetilde{\mathbf{r}})})$.

**Remark 4.7.** For the ease of presentation, let us denote $\mathcal{E} := \|\mathbf{r}-\widetilde{\mathbf{r}}\|_{\mathbf{H}^{-1}}$. If we have $\mathcal{E} = O(1)$, the total regret in Theorem 4.4 becomes $\widetilde{O}(\sqrt{T})$ which matches the regret of linear contextual bandits (Abbasi-Yadkori et al., 2011). We remark that there is a similar assumption in Zhou et al. (2020) where they assume that $\mathbf{r}^\top \mathbf{H}^{-1}\mathbf{r}$ can be upper bounded by a constant. They show that this term can be bounded by the RKHS norm of $\mathbf{r}$ if it belongs to the RKHS induced by the neural tangent kernel (Arora et al., 2019a;b; Lee et al., 2019). In addition, $\mathcal{E}$ here is the difference between the true reward function and the neural network function, which can also be small if the deep neural network function well approximates the reward generating function $r(\cdot)$.

## 5 EXPERIMENTS

In this section, we provide empirical evaluations of Neural-LinUCB on real-world datasets. As we have discussed in Section 3, Neural-LinUCB could be viewed as an instantiation of the Neural-Linear scheme studied in Riquelme et al. (2018) except that we use the UCB exploration instead of the posterior sampling exploration therein. Note that there has been an extensive comparison (Riquelme et al., 2018) of the Neural-Linear methods with many other baselines such as greedy algorithms, Variational Inference, Expectation-Propagation, Bayesian Non-parametrics and so on. Therefore, we do not seek a thorough empirical comparison of Neural-LinUCB with all existing bandits algorithms. In this experiment, we only aim to validate the advantages of our algorithm over the following baselines: (1) Neural-Linear (Riquelme et al., 2018); (2) LinUCB (Chu et al., 2011), which does not have a *deep representation* of the feature vectors; (3) NeuralUCB (Zhou et al., 2020), and (4) NeuralTS (Zhang et al., 2020) which perform UCB/TS exploration on all the parameters of the neural network. All numerical experiments were run on a workstation with Intel(R) Xeon(R) CPU E5-2637 v4 @ 3.50GHz.

**Datasets:** we evaluate the performances of all algorithms on bandit problems created from real-world data. Specifically, following the experimental setting in Zhou et al. (2020),we use datasets *(Shuttle) Statlog*, *Magic* and *Covertype* from UCI machine learning repository (Dua & Graff, 2017), and the *MINST* dataset from LeCun et al. (1998). The details of these datasets are presented in Table 1. In Table 1, each instance represents a feature vector $\mathbf{x} \in \mathbb{R}^d$ that is associated with one of the $K$ arms, and dimension $d$ is the number of attributes in each instance.

**Implementations:** for LinUCB, we follow the setting in Li et al. (2010) to use disjoint models for different arms. For neural network based algorithms, we use a ReLU neural network defined as in (2.3) with $L = 2$ and $m = 100$ for the UCI datasets (*Statlog*, *Magic*, *Covertype*). Thus the neural network weights are $\mathbf{W}_1 \in \mathbb{R}^{m \times d}$, $\mathbf{W}_2 \in \mathbb{R}^{k \times m}$, and $\boldsymbol{\theta} \in \mathbb{R}^k$ respectively, where $k = 100$, $m = 100$, and $d$ is the dimension of features in the corresponding task. Since the problem size of the MNIST dataset is larger, inspired by Hinton & Salakhutdinov (2006), we use a deeper NN and set $L = 3$, $k = 100$ and $m = 100$, with weights $\mathbf{W}_1 \in \mathbb{R}^{m \times d}$, $\mathbf{W}_2 \in \mathbb{R}^{m \times m}$, $\mathbf{W}_3 \in \mathbb{R}^{k \times m}$,

Table 1: Specifications of datasets from the UCI machine learning repository and the MNIST dataset used in this paper.

|                       | *Statlog* | *Magic* | *Covertype* | *MNIST* |
|-----------------------|-----------|---------|-------------|---------|
| Number of attributes  | 9         | 11      | 54          | 784     |
| Number of arms        | 7         | 2       | 7           | 10      |
| Number of instances   | 58,000    | 19,020  | 581,012     | 60,000  |

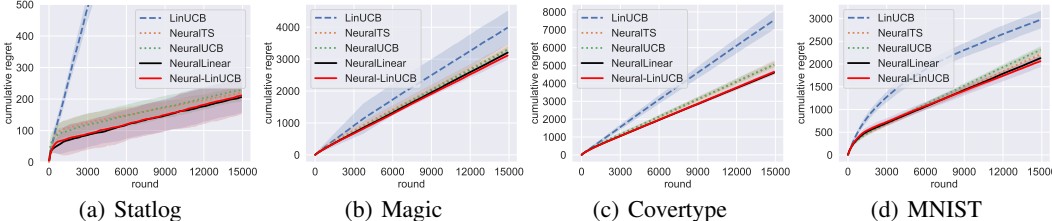

| (a) Statlog | (b) Magic | (c) Covertype | (d) MNIST |

Figure 1: The cumulative regrets of LinUCB, NeuralUCB, Neural-Linear and Neural-LinUCB over $15,000$ rounds. Experiments are averaged over 10 repetitions.

and $\boldsymbol{\theta} \in \mathbb{R}^k$. We set the time horizon $T = 15,000$, which is the total number of rounds for each algorithm on each dataset. We use stochastic gradient decent to optimize the network weights, with a step size $\eta_q = $1e-5 and maximum iteration number $n = 1,000$. To speed up the training process, the network parameter $\mathbf{w}$ is updated every $H = 100$ rounds starting from round 2000. We also apply early stopping when the loss difference of two consecutive iterations is smaller than a threshold of 1e-6. We set $\lambda = 1$ and $\alpha_t = 0.02$ for all algorithms, $t \in [T]$. For NeuralUCB and NeuralTS, since it is computationally unaffordable to perform the original UCB exploration as displayed in Zhou et al. (2020), we follow their experimental setting to replace the matrix $\mathbf{Z}_t \in \mathbb{R}^{(d+\widehat{p}) \times (d+\widehat{p})}$ in their papers with its diagonal matrix.

**Results:** we plot the cumulative regret of all algorithms versus round in Figures 1(a), 1(b) and 1(c) for UCI datasets and in Figure 1(d) for MNIST. The results are reported based on the average of 10 repetitions over different random shuffles of the datasets. It can be seen that algorithms based on neural network representations (NeuralUCB, NeuralTS, Neural-Linear and Neural-LinUCB) consistently outperform the linear contextual bandit method LinUCB, which shows that linear models may lack representation power and find biased estimates for the underlying reward generating function. Furthermore, our proposed Neural-LinUCB achieves a comparable regret with NeuralUCB in all experiments despite the fact that our algorithm only explores in the output layer of the neural network, which is more computationally efficient as we will show in the sequel. The results in our experiment are well aligned with our theory that deep representation and shallow exploration are sufficient to guarantee a good performance of neural contextual bandit algorithms, which is also consistent with the findings in existing literature (Riquelme et al., 2018) that decoupling the representation learning and uncertainty estimation improves the performance.

We also conducted experiments to study the effects of different widths of deep neural networks on the regret performance and to show the computational efficiency of Neural-LinUCB compared with existing neural bandit algorithms. Due to the space limit, we defer the results to Appendix A.

## 6 CONCLUSIONS

In this paper, we propose a new neural contextual bandit algorithm called Neural-LinUCB, which uses the hidden layers of a ReLU neural network as a deep representation of the raw feature vectors and performs UCB type exploration on the last layer of the neural network. By incorporating techniques in liner contextual bandits and neural tangent kernels, we prove that the proposed algorithm achieves a sublinear regret when the width of the network is sufficiently large. This is the first regret analysis of neural contextual bandit algorithms with deep representation and shallow exploration, which have been observed in practice to work well on many benchmark bandit problems (Riquelme et al., 2018). We also conducted experiments on real-world datasets to demonstrate the advantage of the proposed algorithm over LinUCB and existing neural contextual bandit algorithms.

ACKNOWLEDGEMENTS

We thank the anonymous reviewers for their helpful comments. PX and QG are partially supported by the National Science Foundation CAREER Award 1906169 and IIS-1904183. The views and conclusions contained in this paper are those of the authors and should not be interpreted as representing any funding agencies.

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

## A    ADDITIONAL EXPERIMENTAL RESULTS

In this section, we provide more experimental results that are omitted in Section 5 due to space limit.

### A.1    COMPUTATIONAL EFFICIENCY OF NEURAL-LINUCB

Throughout the experiments, our Neural-LinUCB algorithm is much more computationally efficient than NeuralUCB since we only perform the UCB exploration on the last layer of the neural network, where the dimension is much lower. In specific, on the *Statlog* dataset, it takes on average 1.11 seconds for NeuralUCB to finish 100 rounds (one epoch in Algorithm 1) and achieve the regret in Figure 1(a), while it only takes 0.58 seconds for Neural-LinUCB to finish 100 rounds and achieve the comparable or even better regret in Figure 1(a). On the *Magic* dataset, the average runtimes for 100 rounds of NeuralUCB and Neural-LinUCB are 1.32 seconds and 0.81 seconds respectively. On the *Covertype* dataset, the runtimes of NeuralUCB and Neural-LinUCB are 1.02 seconds and 0.66 seconds respectively. And on the MNIST dataset, the average runtimes for 100 rounds of NeuralUCB and Neural-LinUCB are 4.67 seconds and 1.29 seconds respectively. For practical applications in the real-world with larger problem sizes, we believe that the improvement of our algorithm in terms of the computational efficiency will be more pronounced.

As we discussed in Section 5 and in the above paragraph as well, the computational efficiency of Neural-LinUCB mainly stems from the design of shallow exploration. This is because in UCB based bandit algorithms we need to compute the inverse of matrix $A$ at every time step for arm selection (Line 5 of Algorithm 1). Due to the large width of the neural network used in practice, the arm selection operation could be rather time consuming. However, the neural network weight can be updated periodically (i.e., in our paper it is only updated every $H$ steps). To validate our analysis on computational efficiency, we further studied the time profiling of the experiments conducted on MNIST to compared our proposed algorithm with NeuralUCB in more details.

Table 2: Profiling experiment on MNIST for running 100 rounds: runtime (seconds) for different algorithms on arm selection and network weight update.

| Operations | NeuralUCB | Neural-LinUCB |
|---|---|---|
| Arm selection (Line 5 in Algorithm 1) | 3.60 | 0.28 |
| Network weight update (Line 9 in Algorithm 1) | 0.96 | 0.92 |

In particular, the setting is the same as that in Section 5 for MNIST experiments. We record the time cost of the most expensive two subroutines: (1) the operation of arm selection (Line 5 in Algorithm 1); and (2) the operation of updating the neural network weights (Line 9 in Algorithm 1), for $H = 100$ rounds. The time cost is presented in Table 2. For Neural-LinUCB, the arm selection operation takes about $0.28$ seconds (this is $21.71\%$ of the total time cost by the algorithm in these $H = 100$ rounds), among which the matrix inverse step only takes $0.17$ seconds. For NeuralUCB, the arm selection operation takes $3.60$ seconds (this is $77.19\%$ of the total time time cost by NeuralUCB for $H = 100$ rounds). Therefore, the operation of arm selection in NeuralUCB is much (almost 13 times) more time consuming than that in Neural-LinUCB. Moreover, since the UCB matrix $\mathbf{Z}_t$ in NeuralUCB is defined as $\nabla f(\mathbf{x}; \mathbf{w}) \nabla f(\mathbf{x}; \mathbf{w})^\top$, it needs to compute the gradients via back-propagation ($0.93$ seconds) and compute the matrix inverse ($1.54$ seconds), while our Neural-LinUCB algorithm only needs to compute the matrix inverse of a small matrix ($0.17$ seconds). To summarize, our method is much more computationally efficient.

### A.2    IMPACT OF LARGE WIDTHS

Note that the requirement of width $m$ in our Theorem 4.4 is extremely high. On one hand, our theory may be too conservative since the current understanding of deep learning is still very limited in the field. We believe our work is a good starting point towards understanding the behavior of deep bandits algorithms. On the other hand, we would also like to investigate the impact of mild overparameterization on the regret performance of Neural-LinUCB in practice. Therefore, we conducted additional experiments on the *Statlog* dataset with wider neural networks. In particular, the neural

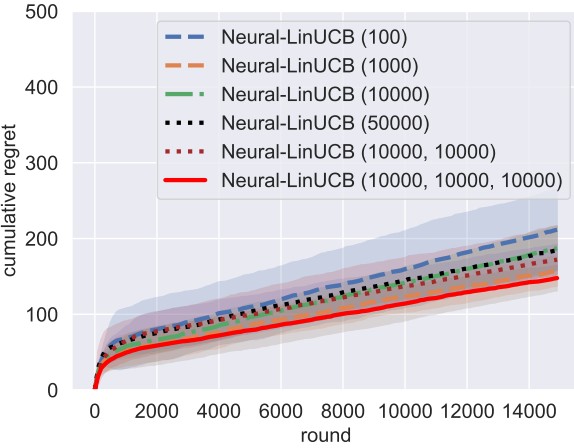

Figure 2: Performance of Neural-LinUCB with different widths on *Statlog* dataset.

network parameters are listed as follows

$$\mathbf{W}_1 \in \mathbb{R}^{m \times d}, \mathbf{W}_2 \in \mathbb{R}^{m \times m}, \dots, \mathbf{W}_L \in \mathbb{R}^{k \times m}, \boldsymbol{\theta} \in \mathbb{R}^k,$$

where $L$ is the depth, $k = 100$, $d$ is the feature dimensions, and $m$ is the width. We conducted experiments for the following settings: (1) $L = 2$, $m = 100$, and thus the hidden layer width is (100); (2) $L = 2$, $m = 1000$, and thus the hidden layer width is (1000); (3) $L = 2$, $m = 10000$, and thus the hidden layer width is (10000); (4) $L = 2$, and the hidden layer width is (50000); (5) $L = 3$, $m = 10000$, and thus the hidden layer width is (10000, 10000); and (6) $L = 4$, $m = 10000$, and thus the hidden layer width is (10000, 10000, 10000). The results are plotted in Figure A.2. We observe that the performance of our Neural-LinUCB algorithm is not negatively impacted by the width of the neural network. In fact, Figure A.2 shows improved performance of Neural-LinUCB when the total number of hidden nodes increases. This is consistent to the observations in Zhang et al. (2017) that an overparameterized neural network trained by gradient descent does not necessarily lead to overfitting and also aligns with our Theorem 4.4 that the regret bound of Neural-LinUCB decreases as the width $m$ increases.

## B  MORE DISCUSSIONS ON ASSUMPTION 4.2

In this section, we are going to show that Assumption 4.2 could be satisfied as long as the feature vectors $\{\mathbf{x}\}$ lie in a begin subspace of $\mathbb{R}^d$. Let us start with the case that $\phi : \mathbb{R}^d \rightarrow \mathbb{R}^m$ is a two-layer ReLU neural network with vector output. In particular, we define $\phi(\mathbf{x}; \mathbf{w})$ as follows $\phi(\mathbf{x}; \mathbf{w}) = \sigma(\mathbf{W}_2 \sigma(\mathbf{W}_1 \mathbf{x}))$, where $\mathbf{w} = (\text{vec}(\mathbf{W}_1), \text{vec}(\mathbf{W}_2))^\top$, $\mathbf{W}_1 \in \mathbb{R}^{m \times d}$, $\mathbf{W}_2 \in \mathbb{R}^{d \times m}$, and $\sigma$ is the ReLU activation function applied elementwise. We use $\mathbf{u}_i^\top$ to denote the $i$-th row of $\mathbf{W}_1$ and thus $\mathbf{W}_1 = (\mathbf{u}_1, \dots, \mathbf{u}_m)^\top$, where $\mathbf{u}_i \in \mathbb{R}^d$, $\forall i \in [m]$. Similarly, we have $\mathbf{W}_2 = (\mathbf{v}_1, \dots, \mathbf{v}_d)^\top$, where $\mathbf{v}_j \in \mathbb{R}^m$ is the $j$-th row of $\mathbf{W}_2$, $\forall j \in [d]$. Let us denote $\mathbf{h}$ as the vector $\sigma(\mathbf{W}_1 \mathbf{x})$. We thus obtain

$$\mathbf{h} = \begin{bmatrix} \mathbb{1}\{\mathbf{u}_1^\top \mathbf{x} \geq 0\} \mathbf{u}_1^\top \mathbf{x} \\ \vdots \\ \mathbb{1}\{\mathbf{u}_m^\top \mathbf{x} \geq 0\} \mathbf{u}_m^\top \mathbf{x} \end{bmatrix}, \quad \phi(\mathbf{x}; \mathbf{w}) = \begin{bmatrix} \mathbb{1}\{\mathbf{v}_1^\top \mathbf{h} \geq 0\} \mathbf{v}_1^\top \mathbf{h} \\ \vdots \\ \mathbb{1}\{\mathbf{v}_d^\top \mathbf{h} \geq 0\} \mathbf{v}_d^\top \mathbf{h} \end{bmatrix}.$$

We use $\phi_l(\mathbf{x}; \mathbf{w})$ to denote the $l$-th entry of vector $\phi(\mathbf{x}; \mathbf{w})$, for any $l \in [d]$. Then it holds that

$$\frac{\partial \phi_l(\mathbf{x}; \mathbf{w})}{\partial \text{vec}(\mathbf{W}_1)} = \mathbb{1}\{\mathbf{v}_1^\top \mathbf{h} \geq 0\} \left[ v_1^1 \mathbb{1}\{\mathbf{u}_1^\top \mathbf{x} \geq 0\} \mathbf{x}^\top, \dots, v_1^m \mathbb{1}\{\mathbf{u}_m^\top \mathbf{x} \geq 0\} \mathbf{x}^\top \right],$$

for all $l \in [d]$, where $v_1^i$ is the $i$-th element in $\mathbf{v}_1$, $i \in [m]$. This further implies that

$$\frac{\partial \phi(\mathbf{x}; \mathbf{w})}{\partial \text{vec}(\mathbf{W}_1)} = \begin{bmatrix} \mathbb{1}\{\mathbf{v}_1^\top \mathbf{h} \geq 0\} \left( v_1^1 \mathbb{1}\{\mathbf{u}_1^\top \mathbf{x} \geq 0\} \mathbf{x}^\top, \dots, v_1^m \mathbb{1}\{\mathbf{u}_m^\top \mathbf{x} \geq 0\} \mathbf{x}^\top \right) \\ \vdots \\ \mathbb{1}\{\mathbf{v}_d^\top \mathbf{h} \geq 0\} \left( v_d^1 \mathbb{1}\{\mathbf{u}_1^\top \mathbf{x} \geq 0\} \mathbf{x}^\top, \dots, v_d^m \mathbb{1}\{\mathbf{u}_m^\top \mathbf{x} \geq 0\} \mathbf{x}^\top \right) \end{bmatrix} \in \mathbb{R}^{d \times md}.$$

Similarly, we can compute the gradient of $\phi(\mathbf{x}; \mathbf{w})$ with respect to $\mathbf{W}_2$. In particular, we have

$$\frac{\partial \phi_1(\mathbf{x}; \mathbf{w})}{\partial \text{vec}(\mathbf{v}_1)} = \mathbb{1}\{\mathbf{v}_1^\top \mathbf{h} \geq 0\} \big[ \mathbb{1}\{\mathbf{u}_1^\top \mathbf{x} \geq 0\} \mathbf{u}_1^\top \mathbf{x}, \ldots, \mathbb{1}\{\mathbf{u}_m^\top \mathbf{x} \geq 0\} \mathbf{u}_m^\top \mathbf{x} \big],$$

$$\frac{\partial \phi_1(\mathbf{x}; \mathbf{w})}{\partial \text{vec}(\mathbf{v}_j)} = [0, \ldots, 0], \quad j \neq 1.$$

Therefore, the gradient of $\phi(\mathbf{x}; \mathbf{w})$ with respect to $\mathbf{W}_2$ is

$$\frac{\partial \phi(\mathbf{x}; \mathbf{w})}{\partial \text{vec}(\mathbf{W}_2)} = \begin{bmatrix} \frac{\partial \phi_1(\mathbf{x}; \mathbf{w})}{\partial \text{vec}(\mathbf{v}_1)} & \cdots & \mathbf{0}^\top \\ & \ddots & \\ \mathbf{0}^\top & \cdots & \frac{\partial \phi_d(\mathbf{x}; \mathbf{w})}{\partial \text{vec}(\mathbf{v}_d)} \end{bmatrix} \in \mathbb{R}^{d \times md}.$$

Lastly, we have

$$\frac{\partial \phi(\mathbf{x}; \mathbf{w})}{\partial \mathbf{w}} = \begin{bmatrix} \frac{\partial \phi(\mathbf{x}; \mathbf{w})}{\partial \text{vec}(\mathbf{W}_1)} & \frac{\partial \phi(\mathbf{x}; \mathbf{w})}{\partial \text{vec}(\mathbf{W}_2)} \end{bmatrix} \in \mathbb{R}^{d \times (md+md)}.$$

Therefore, for any two feature vectors $\mathbf{x}$ and $\mathbf{x}'$ from $\{\mathbf{x}_{i,k}\}_{i \in [T], k \in [K]}$, if many nodes in the initial neural network $\phi(\mathbf{x}; \mathbf{w}_0)$ are activated or deactivated at the same time for both $\mathbf{x}$ and $\mathbf{x}'$, then the spectral norm of the matrix $\frac{\partial \phi(\mathbf{x}; \mathbf{w}_0)}{\partial \mathbf{w}} - \frac{\partial \phi(\mathbf{x}'; \mathbf{w}_0)}{\partial \mathbf{w}}$ would satisfy the condition in Assumption 4.2. A more thorough study of this stability condition is out of the scope of this paper, though it would be an interesting open direction in the theory of deep neural networks.

## C  PROOF OF THE MAIN RESULTS

In this section, we provide the proof of the regret bound for Neural-LinUCB. Recall that in neural contextual bandits, we do not assume a specific formulation of the underlying reward generating function $r(\cdot)$. Instead, we use deep neural networks defined in Section 2.2 to approximate $r(\cdot)$. We will first show that the reward generating function $r(\cdot)$ can be approximated by the local linearization of the overparameterized neural network near the initialization weight $\mathbf{w}^{(0)}$. In particular, we denote the gradient of $\phi(\mathbf{x}; \mathbf{w})$ with respect to $\mathbf{w}$ by $\mathbf{g}(\mathbf{x}; \mathbf{w})$, namely,

$$\mathbf{g}(\mathbf{x}; \mathbf{w}) = \nabla_{\mathbf{w}} \phi(\mathbf{x}; \mathbf{w}), \tag{C.1}$$

which is a matrix in $\mathbb{R}^{d \times p}$. We define $\phi_j(\mathbf{x}; \mathbf{w})$ to be the $j$-th entry of vector $\phi(\mathbf{x}; \mathbf{w})$, for any $j \in [d]$. Then, we can prove the following lemma.

**Lemma C.1.** Suppose Assumptions 4.3 hold. Then there exists $\mathbf{w}^* \in \mathbb{R}^p$ such that $\|\mathbf{w}^* - \mathbf{w}^{(0)}\|_2 \leq 1/\sqrt{m} \sqrt{(\mathbf{r} - \widetilde{\mathbf{r}})^\top \mathbf{H}^{-1}(\mathbf{r} - \widetilde{\mathbf{r}})}$ and it holds that

$$r(\mathbf{x}_{t,k}) = \boldsymbol{\theta}^{*\top} \phi(\mathbf{x}_{t,k}; \mathbf{w}_{t-1}) + \boldsymbol{\theta}_0^\top \mathbf{g}(\mathbf{x}_{t,k}; \mathbf{w}^{(0)})(\mathbf{w}^* - \mathbf{w}^{(0)}),$$

for all $k \in [K]$ and $t = 1, \ldots, T$.

Lemma C.1 implies that the reward generating function $r(\cdot)$ at points $\{\mathbf{x}_{i,k}\}_{i \in [T], k \in [K]}$ can be approximated by a linear function around the initial point $\mathbf{w}^{(0)}$. Note that a similar lemma is also proved in Zhou et al. (2020) for NeuralUCB.

The next lemma shows the upper bounds of the output of the neural network $\phi$ and its gradient.

**Lemma C.2.** Suppose Assumptions 4.1 and 4.3 hold. For any round index $t \in [T]$, suppose it is in the $q$-th epoch of Algorithm 2, i.e., $t = (q-1)H + i$ for some $i \in [H]$. If the step size $\eta_q$ in Algorithm 2 satisfies

$$\eta \leq \frac{C_0}{d^2 m n T^{5.5} L^6 \log(TK/\delta)},$$

and the width of the neural network satisfies

$$m \geq \max\{L \log(TK/\delta), dL^2 \log(m/\delta), \delta^{-6} H^{18} L^{16} \log^3(TK)\}, \tag{C.2}$$

then, with probability at least $1 - \delta$ we have

$$\|\mathbf{w}_t - \mathbf{w}^{(0)}\|_2 \leq \frac{\delta^{3/2}}{m^{1/2}Tn^{9/2}L^6\log^3(m)},$$

$$\|\mathbf{g}(\mathbf{x}_{t,k}; \mathbf{w}^{(0)})\|_F \leq C_1\sqrt{dLm},$$

$$\|\boldsymbol{\phi}(\mathbf{x}; \mathbf{w}_t)\|_2 \leq \sqrt{d\log(n)\log(TK/\delta)},$$

for all $t \in [T]$, $k \in [K]$, where the neural network $\phi$ is defined in (2.4) and its gradient is defined in (C.1).

The next lemma shows that the neural network $\phi(\mathbf{x}; \mathbf{w})$ is close to a linear function in terms of the weight $\mathbf{w}$ parameter around a small neighborhood of the initialization point $\mathbf{w}^{(0)}$.

**Lemma C.3** (Theorems 5 in Cao & Gu (2019b))**.** Let $\mathbf{w}, \mathbf{w}'$ be in the neighborhood of $\mathbf{w}_0$, i.e., $\mathbf{w}, \mathbf{w}' \in \mathbb{B}(\mathbf{w}_0, \omega)$ for some $\omega > 0$. Consider the neural network defined in (2.4), if the width $m$ and the radius $\omega$ of the neighborhood satisfy

$$m \geq C_0\max\{dL^2\log(m/\delta), \omega^{-4/3}L^{-8/3}\log(TK)\log(m/(\omega\delta))\},$$

$$\omega \leq C_1 L^{-5}(\log m)^{-3/2},$$

then for all $\mathbf{x} \in \{\mathbf{x}_{t,k}\}_{t \in [T], k \in [K]}$, with probability at least $1 - \delta$ it holds that

$$|\phi_j(\mathbf{x}; \mathbf{w}) - \widehat{\phi}_j(\mathbf{x}; \mathbf{w})| \leq C_2\omega^{4/3}L^3 d^{-1/2}\sqrt{m\log m},$$

where $\widehat{\phi}_j(\mathbf{x}; \mathbf{w})$ is the linearization of $\phi_j(\mathbf{x}; \mathbf{w})$ at $\mathbf{w}'$ defined as follow:

$$\widehat{\phi}_j(\mathbf{x}; \mathbf{w}) = \phi_j(\mathbf{x}; \mathbf{w}') + \langle\nabla_\mathbf{w}\phi_j(\mathbf{x}; \mathbf{w}'), \mathbf{w} - \mathbf{w}'\rangle. \tag{C.3}$$

Similar results on the local linearization of an overparameterized neural network are also presented in Allen-Zhu et al. (2019b); Cao & Gu (2019b).

For the output layer $\boldsymbol{\theta}^*$, we perform a UCB type exploration and thus we need to characterize the uncertainty of the estimation. The next lemma shows the confidence bound of the estimate $\boldsymbol{\theta}_t$ in Algorithm 1.

**Lemma C.4.** Suppose Assumption and 4.3 hold. For any $\delta \in (0, 1)$, with probability at least $1 - \delta$, the distance between the estimated weight vector $\boldsymbol{\theta}_t$ by Algorithm 1 and $\boldsymbol{\theta}^*$ can be bounded as follows:

$$\left\|\boldsymbol{\theta}_t - \boldsymbol{\theta}^* - \mathbf{A}_t^{-1}\sum_{s=1}^t \boldsymbol{\phi}(\mathbf{x}_{s,a_s}; \mathbf{w}_{s-1})\boldsymbol{\theta}_0^\top \mathbf{g}(\mathbf{x}_{s,a_s}; \mathbf{w}^{(0)})(\mathbf{w}^* - \mathbf{w}^{(0)})\right\|_{\mathbf{A}_t}$$

$$\leq \nu\sqrt{2\big(d\log(1 + t(\log HK)/\lambda) + \log 1/\delta\big)} + \lambda^{1/2}M,$$

for any $t \in [T]$.

Note that the confidence bound in Lemma C.4 is different from the standard result for linear contextual bandits in Abbasi-Yadkori et al. (2011). The additional term on the left hand side of the confidence bound is due to the bias caused by the representation learning using a deep neural network. To deal with this extra term, we need the following technical lemma.

**Lemma C.5.** Assume that $\mathbf{A}_t = \lambda\mathbf{I} + \sum_{s=1}^t \boldsymbol{\phi}_s\boldsymbol{\phi}_s^\top$, where $\boldsymbol{\phi}_t \in \mathbb{R}^d$ and $\|\boldsymbol{\phi}_t\|_2 \leq G$ for all $t \geq 1$ and some constants $\lambda, G > 0$. Let $\{\zeta_t\}_{t=1,\ldots}$ be a real-value sequence such that $|\zeta_t| \leq U$ for some constant $U > 0$. Then we have

$$\left\|\mathbf{A}_t^{-1}\sum_{s=1}^t \boldsymbol{\phi}_s\zeta_s\right\|_2 \leq 2Ud, \quad \forall t = 1, 2, \ldots$$

The next lemma provides some standard bounds on the feature matrix $\mathbf{A}_t$, which is a combination of Lemma 10 and Lemma 11 in Abbasi-Yadkori et al. (2011).

**Lemma C.6.** Let $\{\mathbf{x}_t\}_{t=1}^\infty$ be a sequence in $\mathbb{R}^d$ and $\lambda > 0$. Suppose $\|\mathbf{x}_t\|_2 \leq G$ and $\lambda \geq \max\{1, G^2\}$ for some $G > 0$. Let $\mathbf{A}_t = \lambda \mathbf{I} + \sum_{s=1}^t \mathbf{x}_t \mathbf{x}_t^\top$. Then we have

$$\det(\mathbf{A}_t) \leq (\lambda + tG^2/d)^d, \quad \text{and} \quad \sum_{t=1}^T \|\mathbf{x}_t\|_{\mathbf{A}_{t-1}^{-1}}^2 \leq 2\log\frac{\det(\mathbf{A}_T)}{\det(\lambda\mathbf{I})} \leq 2d\log(1 + TG^2/(\lambda d)).$$

Now we are ready to prove the regret bound of Algorithm 1.

*Proof of Theorem 4.4.* For a time horizon $T$, without loss of generality, we assume $T = QH$ for some epoch number $Q$. By the definition of regret in (4.3), we have

$$R_T = \mathbb{E}\bigg[\sum_{t=1}^T (\widehat{r}(\mathbf{x}_{t,a_t^*}) - \widehat{r}(\mathbf{x}_{t,a_t}))\bigg] = \mathbb{E}\bigg[\sum_{q=1}^Q \sum_{i=1}^H (\widehat{r}(\mathbf{x}_{qH+i,a_{qH+i}^*}) - \widehat{r}(\mathbf{x}_{qH+i,a_{qH+i}}))\bigg].$$

Note that for the simplicity of presentation, we omit the conditional expectation notation of $\mathbf{w}^{(0)}$ in the rest of the proof when the context is clear. In the second equation, we rewrite the time index $t = qH + i$ as the $i$-th iteration in the $q$-th epoch.

By the definition in (2.1), we have $\mathbb{E}[\widehat{r}(\mathbf{x}_{t,k})|\mathbf{x}_{t,k}] = r(\mathbf{x}_{t,k})$ for all $t \in [T]$ and $k \in K$. Based on the linearization of reward generating function, we can decompose the instaneous regret into different parts and upper bound them individually. In particular, by Lemma C.1, there exists a vector $\mathbf{w}^* \in \mathbb{R}^p$ such that we can write the expectation of the reward generating function as a linear function. Then it holds that

$$\begin{aligned}
r(\mathbf{x}_{t,a_t^*}) - r(\mathbf{x}_{t,a_t}) &= \boldsymbol{\theta}_0^\top \big[\mathbf{g}(\mathbf{x}_{t,a_t^*}; \mathbf{w}^{(0)}) - \mathbf{g}(\mathbf{x}_{t,a_t}; \mathbf{w}^{(0)})\big](\mathbf{w}^* - \mathbf{w}^{(0)}) \\
&\quad + \boldsymbol{\theta}^{*\top}\big[\boldsymbol{\phi}(\mathbf{x}_{t,a_t^*}; \mathbf{w}_{t-1}) - \boldsymbol{\phi}(\mathbf{x}_{t,a_t}; \mathbf{w}_{t-1})\big] \\
&= \boldsymbol{\theta}_0^\top\big[\mathbf{g}(\mathbf{x}_{t,a_t^*}; \mathbf{w}^{(0)}) - \mathbf{g}(\mathbf{x}_{t,a_t}; \mathbf{w}^{(0)})\big](\mathbf{w}^* - \mathbf{w}^{(0)}) \\
&\quad + \boldsymbol{\theta}_{t-1}^\top\big[\boldsymbol{\phi}(\mathbf{x}_{t,a_t^*}; \mathbf{w}_{t-1}) - \boldsymbol{\phi}(\mathbf{x}_{t,a_t}; \mathbf{w}_{t-1})\big] \\
&\quad - (\boldsymbol{\theta}_{t-1} - \boldsymbol{\theta}^*)^\top\big[\boldsymbol{\phi}(\mathbf{x}_{t,a_t^*}; \mathbf{w}_{t-1}) - \boldsymbol{\phi}(\mathbf{x}_{t,a_t}; \mathbf{w}_{t-1})\big]. \quad\quad (\text{C}.4)
\end{aligned}$$

The first term in (C.4) can be easily bounded using the first order stability in Assumption 4.2 and the distance between $\mathbf{w}^*$ and $\mathbf{w}^{(0)}$ in Lemma C.1. The second term in (C.4) is related to the optimistic rule of choosing arms in Line 5 of Algorithm 1, which can be bounded using the same technique for LinUCB (Abbasi-Yadkori et al., 2011). For the last term in (C.4), we need to prove that the estimate of weight parameter $\boldsymbol{\theta}_{t-1}$ lies in a confidence ball centered at $\boldsymbol{\theta}^*$. For the ease of notation, we define

$$\mathbf{M}_t = \mathbf{A}_t^{-1}\sum_{s=1}^t \boldsymbol{\phi}(\mathbf{x}_{s,a_s}; \mathbf{w}_{s-1})\boldsymbol{\theta}_0^\top \mathbf{g}(\mathbf{x}_{s,a_s}; \mathbf{w}^{(0)})(\mathbf{w}^* - \mathbf{w}^{(0)}). \quad\quad (\text{C}.5)$$

Then the second term in (C.4) can be bounded in the following way:

$$\begin{aligned}
&-(\boldsymbol{\theta}_{t-1} - \boldsymbol{\theta}^*)^\top\big[\boldsymbol{\phi}(\mathbf{x}_{t,a_t^*}; \mathbf{w}_{t-1}) - \boldsymbol{\phi}(\mathbf{x}_{t,a_t}; \mathbf{w}_{t-1})\big] \\
&= -(\boldsymbol{\theta}_{t-1} - \boldsymbol{\theta}^* - \mathbf{M}_{t-1})^\top\boldsymbol{\phi}(\mathbf{x}_{t,a_t^*}; \mathbf{w}_{t-1}) + (\boldsymbol{\theta}_{t-1} - \boldsymbol{\theta}^* - \mathbf{M}_{t-1})^\top\boldsymbol{\phi}(\mathbf{x}_{t,a_t}; \mathbf{w}_{t-1}) \\
&\quad - \mathbf{M}_{t-1}^\top\big[\boldsymbol{\phi}(\mathbf{x}_{t,a_t^*}; \mathbf{w}_{t-1}) - \boldsymbol{\phi}(\mathbf{x}_{t,a_t}; \mathbf{w}_{t-1})\big] \\
&\leq \|\boldsymbol{\theta}_{t-1} - \boldsymbol{\theta}^* - \mathbf{M}_{t-1}\|_{\mathbf{A}_{t-1}} \cdot \|\boldsymbol{\phi}(\mathbf{x}_{t,a_t^*}; \mathbf{w}_{t-1})\|_{\mathbf{A}_{t-1}^{-1}} \\
&\quad + \|\boldsymbol{\theta}_{t-1} - \boldsymbol{\theta}^* - \mathbf{M}_{t-1}\|_{\mathbf{A}_{t-1}} \cdot \|\boldsymbol{\phi}(\mathbf{x}_{t,a_t}; \mathbf{w}_{t-1})\|_{\mathbf{A}_{t-1}^{-1}} \\
&\quad + \big\|\mathbf{M}_{t-1}^\top\big[\boldsymbol{\phi}(\mathbf{x}_{t,a_t^*}; \mathbf{w}_{t-1}) - \boldsymbol{\phi}(\mathbf{x}_{t,a_t}; \mathbf{w}_{t-1})\big]\big\|_2 \\
&\leq \alpha_t\|\boldsymbol{\phi}(\mathbf{x}_{t,a_t^*}; \mathbf{w}_{t-1})\|_{\mathbf{A}_{t-1}^{-1}} + \alpha_t\|\boldsymbol{\phi}(\mathbf{x}_{t,a_t}; \mathbf{w}_{t-1})\|_{\mathbf{A}_{t-1}^{-1}} \\
&\quad + \|\mathbf{M}_{t-1}\|_2 \cdot \|\boldsymbol{\phi}(\mathbf{x}_{t,a_t^*}; \mathbf{w}_{t-1}) - \boldsymbol{\phi}(\mathbf{x}_{t,a_t}; \mathbf{w}_{t-1})\|_2. \quad\quad (\text{C}.6)
\end{aligned}$$

where the last inequality is due to Lemma C.4 and the choice of $\alpha_t$. Plugging (C.6) back into (C.4) yields

$$
\begin{aligned}
r(\mathbf{x}_{t,a_t^*}) - r(\mathbf{x}_{t,a_t}) \leq\ & \alpha_t \|\boldsymbol{\phi}(\mathbf{x}_{t,a_t};\mathbf{w}_{t-1})\|_{\mathbf{A}_{t-1}^{-1}} - \alpha_t \|\boldsymbol{\phi}(\mathbf{x}_{t,a_t^*};\mathbf{w}_{t-1})\|_{\mathbf{A}_{t-1}^{-1}} \\
& + \alpha_t \|\boldsymbol{\phi}(\mathbf{x}_{t,a_t^*};\mathbf{w}_{t-1})\|_{\mathbf{A}_{t-1}^{-1}} + \alpha_t \|\boldsymbol{\phi}(\mathbf{x}_{t,a_t};\mathbf{w}_{t-1})\|_{\mathbf{A}_{t-1}^{-1}} \\
& + \|\mathbf{M}_{t-1}\|_2 \cdot \|\boldsymbol{\phi}(\mathbf{x}_{t,a_t^*};\mathbf{w}_{t-1}) - \boldsymbol{\phi}(\mathbf{x}_{t,a_t};\mathbf{w}_{t-1})\|_2 \\
& + \|\boldsymbol{\theta}_0\|_2 \cdot \|\mathbf{g}(\mathbf{x}_{t,a_t^*};\mathbf{w}^{(0)}) - \mathbf{g}(\mathbf{x}_{t,a_t};\mathbf{w}^{(0)})\|_F \cdot \|\mathbf{w}^* - \mathbf{w}^{(0)}\|_2 \\
\leq\ & 2\alpha_t \|\boldsymbol{\phi}(\mathbf{x}_{t,a_t};\mathbf{w}_{t-1})\|_{\mathbf{A}_{t-1}^{-1}} + \|\mathbf{M}_{t-1}\|_2 \cdot \|\boldsymbol{\phi}(\mathbf{x}_{t,a_t^*};\mathbf{w}_{t-1}) - \boldsymbol{\phi}(\mathbf{x}_{t,a_t};\mathbf{w}_{t-1})\|_2 \\
& + \ell_{\mathrm{Lip}}\|\boldsymbol{\theta}_0\|_2 \cdot \|\mathbf{x}_{t,a_t^*} - \mathbf{x}_{t,a_t}\|_2 \cdot \|\mathbf{w}^* - \mathbf{w}^{(0)}\|_2,
\end{aligned}
\tag{C.7}
$$

where in the first inequality we used the definition of upper confidence bound in Algorithm 1 and the second inequality is due to Assumption 4.2. Recall the linearization of $\phi_j$ in Lemma C.3, we have

$$
\widehat{\boldsymbol{\phi}}(\mathbf{x};\mathbf{w}_{t-1}) = \boldsymbol{\phi}(\mathbf{x};\mathbf{w}_0) + \mathbf{g}(\mathbf{x};\mathbf{w}_0)(\mathbf{w}_{t-1} - \mathbf{w}_0).
$$

Note that by the initialization, we have $\boldsymbol{\phi}(\mathbf{x};\mathbf{w}_0) = \mathbf{0}$ for any $\mathbf{x} \in \mathbb{R}^d$. Thus, it holds that

$$
\begin{aligned}
& \boldsymbol{\phi}(\mathbf{x}_{t,a_t^*};\mathbf{w}_{t-1}) - \boldsymbol{\phi}(\mathbf{x}_{t,a_t};\mathbf{w}_{t-1}) \\
& = \boldsymbol{\phi}(\mathbf{x}_{t,a_t^*};\mathbf{w}_{t-1}) - \boldsymbol{\phi}(\mathbf{x}_{t,a_t^*};\mathbf{w}_0) + \boldsymbol{\phi}(\mathbf{x}_{t,a_t};\mathbf{w}_0) - \boldsymbol{\phi}(\mathbf{x}_{t,a_t};\mathbf{w}_{t-1}) \\
& = \boldsymbol{\phi}(\mathbf{x}_{t,a_t^*};\mathbf{w}_{t-1}) - \widehat{\boldsymbol{\phi}}(\mathbf{x}_{t,a_t^*};\mathbf{w}_{t-1}) + \mathbf{g}(\mathbf{x}_{t,a_t^*};\mathbf{w}_0)(\mathbf{w}_{t-1} - \mathbf{w}_0) \\
& \quad + \boldsymbol{\phi}(\mathbf{x}_{t,a_t};\mathbf{w}_{t-1}) - \widehat{\boldsymbol{\phi}}(\mathbf{x}_{t,a_t};\mathbf{w}_{t-1}) - \mathbf{g}(\mathbf{x}_{t,a_t};\mathbf{w}_0)(\mathbf{w}_{t-1} - \mathbf{w}_0),
\end{aligned}
\tag{C.8}
$$

which immediately implies that

$$
\begin{aligned}
& \big\|\boldsymbol{\phi}(\mathbf{x}_{t,a_t^*};\mathbf{w}_{t-1}) - \boldsymbol{\phi}(\mathbf{x}_{t,a_t};\mathbf{w}_{t-1})\big\|_2 \\
& \leq \big\|\boldsymbol{\phi}(\mathbf{x}_{t,a_t^*};\mathbf{w}_{t-1}) - \widehat{\boldsymbol{\phi}}(\mathbf{x}_{t,a_t^*};\mathbf{w}_{t-1})\big\|_2 + \big\|\boldsymbol{\phi}(\mathbf{x}_{t,a_t};\mathbf{w}_{t-1}) - \widehat{\boldsymbol{\phi}}(\mathbf{x}_{t,a_t};\mathbf{w}_{t-1})\big\|_2 \\
& \quad + \big\|\big(\mathbf{g}(\mathbf{x}_{t,a_t^*};\mathbf{w}_0) - \mathbf{g}(\mathbf{x}_{t,a_t};\mathbf{w}_0)\big)(\mathbf{w}_{t-1} - \mathbf{w}_0)\big\|_2 \\
& \leq C_0 \omega^{4/3} L^3 d^{1/2} \sqrt{m \log m} + \ell_{\mathrm{Lip}}\|\mathbf{x}_{t,a_t^*} - \mathbf{x}_{t,a_t}\|_2 \|\mathbf{w}_{t-1} - \mathbf{w}^{(0)}\|_2,
\end{aligned}
\tag{C.9}
$$

where the second inequality is due to Lemma C.3 and Assumption 4.2. Therefore, the instaneous regret can be further upper bounded as follows.

$$
\begin{aligned}
& r(\mathbf{x}_{t,a_t^*}) - r(\mathbf{x}_{t,a_t}) \\
& \leq 2\alpha_t \|\boldsymbol{\phi}(\mathbf{x}_{t,a_t};\mathbf{w}_{t-1})\|_{\mathbf{A}_{t-1}^{-1}} + \ell_{\mathrm{Lip}}\|\boldsymbol{\theta}_0\|_2 \cdot \|\mathbf{x}_{t,a_t^*} - \mathbf{x}_{t,a_t}\|_2 \cdot \|\mathbf{w}^* - \mathbf{w}^{(0)}\|_2 \\
& \quad + \|\mathbf{M}_{t-1}\|_2 \cdot \big(C_0 \omega^{4/3} L^3 d^{1/2} \sqrt{m \log m} + \ell_{\mathrm{Lip}}\|\mathbf{x}_{t,a_t^*} - \mathbf{x}_{t,a_t}\|_2 \|\mathbf{w}_{t-1} - \mathbf{w}^{(0)}\|_2\big).
\end{aligned}
\tag{C.10}
$$

By Assumption 4.1 we have $\|\mathbf{x}_{t,a_t^*} - \mathbf{x}_{t,a_t}\|_2 \leq 2$. By Lemma C.1 and Lemma C.2, we have

$$
\begin{aligned}
\|\mathbf{w}^* - \mathbf{w}^{(0)}\|_2 &\leq \sqrt{1/m(\mathbf{r} - \widetilde{\mathbf{r}})^\top \mathbf{H}^{-1}(\mathbf{r} - \widetilde{\mathbf{r}})}, \\
\|\mathbf{w}_t - \mathbf{w}^{(0)}\|_2 &\leq \frac{\delta^{3/2}}{m^{1/2} T n^{9/2} L^6 \log^3(m)}.
\end{aligned}
\tag{C.11}
$$

In addition, since the entries of $\boldsymbol{\theta}_0$ are i.i.d. generated from $N(0, 1/d)$, we have $\|\boldsymbol{\theta}_0\|_2 \leq 2(2 + \sqrt{d^{-1}\log(1/\delta)})$ with probability at least $1 - \delta$ for any $\delta > 0$. By Lemma C.2, we have $\|\mathbf{g}(\mathbf{x}_{t,a_t};\mathbf{w}^{(0)})\|_F \leq C_1\sqrt{dm}$. Therefore,

$$
\big|\boldsymbol{\theta}_0^\top \mathbf{g}(\mathbf{x}_{s,a_s};\mathbf{w}^{(0)})(\mathbf{w}^* - \mathbf{w}^{(0)})\big| \leq C_2 d\sqrt{\log(1/\delta)(\mathbf{r} - \widetilde{\mathbf{r}})^\top \mathbf{H}^{-1}(\mathbf{r} - \widetilde{\mathbf{r}})}.
$$

Then, by the definition of $\mathbf{M}_t$ in (C.5) and Lemma C.5, we have

$$
\|\mathbf{M}_{t-1}\|_2 \leq C_3 d^2 \sqrt{\log(1/\delta)(\mathbf{r} - \widetilde{\mathbf{r}})^\top \mathbf{H}^{-1}(\mathbf{r} - \widetilde{\mathbf{r}})}.
\tag{C.12}
$$

Substituting (C.12) and the above results on $\|\mathbf{x}_{t,a_t} - \mathbf{x}_{t,a_t^*}\|_2$, $\|\boldsymbol{\theta}_0\|_2$, $\|\mathbf{w}^* - \mathbf{w}^{(0)}\|_2$ and $\|\mathbf{w}_{t-1} - \mathbf{w}^{(0)}\|_2$ back into (C.10) further yields

$$r(\mathbf{x}_{t,a_t^*}) - r(\mathbf{x}_{t,a_t})$$

$$\leq 2\alpha_t \|\boldsymbol{\phi}(\mathbf{x}_{t,a_t}; \mathbf{w}_{t-1})\|_{\mathbf{A}_{t-1}^{-1}} + C_4 \ell_{\text{Lip}} m^{-1/2} \sqrt{\log(1/\delta)(\mathbf{r} - \widetilde{\mathbf{r}})^\top \mathbf{H}^{-1}(\mathbf{r} - \widetilde{\mathbf{r}})}$$

$$+ \left( C_0 \omega^{4/3} L^3 d^{1/2} \sqrt{m \log m} + \frac{2\ell_{\text{Lip}} \delta^{3/2}}{m^{1/2} T n^{9/2} L^6 \log^3(m)} \right) C_3 d^2 \sqrt{\log(1/\delta)(\mathbf{r} - \widetilde{\mathbf{r}})^\top \mathbf{H}^{-1}(\mathbf{r} - \widetilde{\mathbf{r}})}.$$

Note that we have $\omega = O(m^{-1/2}\|\mathbf{r} - \widetilde{\mathbf{r}}\|_{\mathbf{H}^{-1}})$ by Lemma C.1. Therefore, the regret of the Neural-LinUCB is

$$R_T \leq \sqrt{QH \max_{t \in [T]} \alpha_t^2 \sum_{q=1}^{Q} \sum_{i=1}^{H} \|\boldsymbol{\phi}(\mathbf{x}_{i,a_i}; \mathbf{w}_{qH+i})\|_{\mathbf{A}_i^{-1}}^2} + C_4 \ell_{\text{Lip}} m^{-1/2} T \sqrt{\log(1/\delta)} \|\mathbf{r} - \widetilde{\mathbf{r}}\|_{\mathbf{H}^{-1}}$$

$$+ \left( \frac{C_0 T L^3 d^{1/2} \sqrt{\log m} \|\mathbf{r} - \widetilde{\mathbf{r}}\|_{\mathbf{H}^{-1}}^{4/3}}{m^{1/6}} + \frac{2\ell_{\text{Lip}} \delta^{3/2}}{m^{1/2} n^{9/2} L^6 \log^3(m)} \right) C_3 d^2 \sqrt{\log(1/\delta)} \|\mathbf{r} - \widetilde{\mathbf{r}}\|_{\mathbf{H}^{-1}}$$

$$\leq C_5 \sqrt{Td \log(1 + TG^2/(\lambda d))} \left( \nu \sqrt{d \log(1 + T(\log TK)/\lambda) + \log 1/\delta} + \lambda^{1/2} M \right)$$

$$+ C_6 \ell_{\text{Lip}} L^3 d^{5/2} m^{-1/6} T \sqrt{\log m \log(1/\delta) \log(TK/\delta)} \|\mathbf{r} - \widetilde{\mathbf{r}}\|_{\mathbf{H}^{-1}},$$

where the first inequality is due to Cauchy's inequality, the second inequality comes from the upper bound of $\alpha_t$ in Lemma C.4 and Lemma C.6. $\{C_j\}_{j=0,\ldots,6}$ are absolute constants that are independent of problem parameters. $\square$

*Proof of Corollary 4.6.* It directly follows the result in Theorem 4.4. $\square$

# D PROOF OF TECHNICAL LEMMAS

In this section, we provide the proof of technical lemmas used in the regret analysis of Algorithm 1.

## D.1 PROOF OF LEMMA C.1

Before we prove the lemma, we first present some notations and a supporting lemma for simplification. Let $\boldsymbol{\beta} = (\boldsymbol{\theta}^\top, \mathbf{w}^\top)^\top \in \mathbb{R}^{d+p}$ be the concatenation of the exploration parameter and the hidden layer parameter of the neural network $f(\mathbf{x}; \boldsymbol{\beta}) = \boldsymbol{\theta}^\top \boldsymbol{\phi}(\mathbf{x}; \mathbf{w})$. Note that for any input data vector $\mathbf{x} \in \mathbb{R}^d$, we have

$$\frac{\partial}{\partial \boldsymbol{\beta}} f(\mathbf{x}; \boldsymbol{\beta}) = \left( \boldsymbol{\phi}(\mathbf{x}; \mathbf{w})^\top, \boldsymbol{\theta}^\top \frac{\partial}{\partial \mathbf{w}} \boldsymbol{\phi}(\mathbf{x}; \mathbf{w}) \right)^\top = \left( \boldsymbol{\phi}(\mathbf{x}; \mathbf{w})^\top, \boldsymbol{\theta}^\top \mathbf{g}(\mathbf{x}; \mathbf{w}) \right)^\top, \tag{D.1}$$

where $\mathbf{g}(\mathbf{x}; \mathbf{w})$ is the partial gradient of $\boldsymbol{\phi}(\mathbf{x}; \mathbf{w})$ with respect to $\mathbf{w}$ defined in (C.1), which is a matrix in $\mathbb{R}^{d \times p}$. Similar to (4.2), we define $\mathbf{H}_{L+1}$ to be the neural tangent kernel matrix based on all $L + 1$ layers of the neural network $f(\mathbf{x}; \boldsymbol{\beta})$. Note that by the definition of $\mathbf{H}$ in (4.2), we must have $\mathbf{H}_{L+1} = \mathbf{H} + \mathbf{B}$ for some positive definite matrix $\mathbf{B} \in \mathbb{R}^{TK \times TK}$. The following lemma shows that the NTK matrix is close to the matrix defined based on the gradients of the neural network on $TK$ data points.

**Lemma D.1** (Theorem 3.1 in Arora et al. (2019b)). Let $\epsilon > 0$ and $\delta \in (0, 1)$. Suppose the activation function in (2.3) is ReLU, i.e., $\sigma_l(x) = \max(0, x)$, and the width of the neural network satisfies

$$m \geq \boldsymbol{\Omega}\left( \frac{L^{14}}{\epsilon^4} \log\left( \frac{L}{\delta} \right) \right). \tag{D.2}$$

Then for any $\mathbf{x}, \mathbf{x}' \in \mathbb{R}^d$ with $\|\mathbf{x}\|_2 = \|\mathbf{x}'\|_2 = 1$, with probability at least $1 - \delta$ over the randomness of the initialization of the network weight $\mathbf{w}$ it holds that

$$\left| \left\langle \frac{1}{\sqrt{m}} \frac{\partial f(\boldsymbol{\beta}, \mathbf{x})}{\partial \boldsymbol{\beta}}, \frac{1}{\sqrt{m}} \frac{\partial f(\boldsymbol{\beta}, \mathbf{x}')}{\partial \boldsymbol{\beta}} \right\rangle - \mathbf{H}_{L+1}(\mathbf{x}, \mathbf{x}') \right| \leq \epsilon.$$

Note that in the above lemma, there is a factor $1/\sqrt{m}$ before the gradient. This is due to the additional $\sqrt{m}$ factor in the definition of the neural network in (2.3), which ensures the value of the neural network function evaluated at the initialization is of the order $O(1)$.

*Proof of Lemma C.1.* Recall that we renumbered the feature vectors $\{\mathbf{x}_{t,k}\}_{t\in[T],k\in[K]}$ for all arms from round 1 to round $T$ as $\{\mathbf{x}_i\}_{i=1,\dots,TK}$. By concatenating the gradients at different inputs and the gradient in (D.1), we define $\boldsymbol{\Psi} \in \mathbb{R}^{TK\times(d+p)}$ as follows.

$$\boldsymbol{\Psi} = \frac{1}{\sqrt{m}}\begin{bmatrix} \frac{\partial}{\partial\boldsymbol{\beta}}\boldsymbol{\theta}^\top\boldsymbol{\phi}(\mathbf{x}_1;\mathbf{w}) \\ \vdots \\ \frac{\partial}{\partial\boldsymbol{\beta}}\boldsymbol{\theta}^\top\boldsymbol{\phi}(\mathbf{x}_{TK};\mathbf{w}) \end{bmatrix} = \frac{1}{\sqrt{m}}\begin{bmatrix} \boldsymbol{\phi}(\mathbf{x}_1;\mathbf{w}^{(0)})^\top & \boldsymbol{\theta}_0^\top\mathbf{g}(\mathbf{x}_1;\mathbf{w}^{(0)}) \\ \vdots & \vdots \\ \boldsymbol{\phi}(\mathbf{x}_i;\mathbf{w}^{(0)})^\top & \boldsymbol{\theta}_0^\top\mathbf{g}(\mathbf{x}_i;\mathbf{w}^{(0)}) \\ \vdots & \vdots \\ \boldsymbol{\phi}(\mathbf{x}_{TK};\mathbf{w}^{(0)})^\top & \boldsymbol{\theta}_0^\top\mathbf{g}(\mathbf{x}_{TK};\mathbf{w}^{(0)}) \end{bmatrix}.$$

By Applying Lemma D.1, we know with probability at least $1-\delta$ it holds that

$$|\langle\boldsymbol{\Psi}_{j*},\boldsymbol{\Psi}_{l*}\rangle - \mathbf{H}_{L+1}(\mathbf{x}_j,\mathbf{x}_l)| \leq \epsilon$$

for any $\epsilon > 0$ as long as the width $m$ satisfies the condition in (D.2). By applying union bound over all data points $\{\mathbf{x}_1,\dots,\mathbf{x}_t,\dots,\mathbf{x}_{TK}\}$, we further have

$$\|\boldsymbol{\Psi}\boldsymbol{\Psi}^\top - \mathbf{H}_{L+1}\|_F \leq TK\epsilon.$$

Note that $\mathbf{H}$ is the neural tangent kernel (NTK) matrix defined in (4.2) and $\mathbf{H}_{L+1}$ is the NTK matrix defined based on all $L+1$ layers. By Assumption 4.3, $\mathbf{H}$ has a minimum eigenvalue $\lambda_0 > 0$, which is defined based on the first $L$ layers of $f$. Furthermore, by the definition of NTK matrix in (4.2), we know that $\mathbf{H}_{L+1} = \mathbf{H} + \mathbf{B}$ for some semi-positive definite matrix $\mathbf{B}$. Therefore, the NTK matrix $\mathbf{H}_{L+1}$ defined based on all $L+1$ layers is also positive definite and its minimum eigenvalue is lower bounded by $\lambda_0$. Let $\epsilon = \lambda_0/(2TK)$. By triangle equality we have $\boldsymbol{\Psi}\boldsymbol{\Psi}^\top \succ \mathbf{H}_{L+1} - \|\boldsymbol{\Psi}\boldsymbol{\Psi}^\top - \mathbf{H}_{L+1}\|_2\mathbf{I} \succ \mathbf{H}_{L+1} - \|\boldsymbol{\Psi}\boldsymbol{\Psi}^\top - \mathbf{H}_{L+1}\|_F\mathbf{I} \succ \mathbf{H}_{L+1} - \lambda_0/2\mathbf{I} \succ 1/2\mathbf{H}_{L+1}$, which means that $\boldsymbol{\Psi}$ is semi-definite positive and thus $\text{rank}(\boldsymbol{\Psi}) = TK$ since $m > TK$.

We assume that $\boldsymbol{\Psi}$ can be decomposed as $\boldsymbol{\Psi} = \mathbf{P}\mathbf{D}\mathbf{Q}^\top$, where $\mathbf{P} \in \mathbb{R}^{TK\times TK}$ is the eigenvectors of $\boldsymbol{\Psi}\boldsymbol{\Psi}^\top$ and thus $\mathbf{P}\mathbf{P}^\top = \mathbf{I}_{TK}$, $\mathbf{D} \in \mathbb{R}^{TK\times TK}$ is a diagonal matrix with the square root of eigenvalues of $\boldsymbol{\Psi}\boldsymbol{\Psi}^\top$, and $\mathbf{Q}^\top \in \mathbb{R}^{TK\times(d+p)}$ is the eigenvectors of $\boldsymbol{\Psi}^\top\boldsymbol{\Psi}$ and thus $\mathbf{Q}^\top\mathbf{Q} = \mathbf{I}_{TK}$. We use $\mathbf{Q}_1 \in \mathbb{R}^{d\times TK}$ and $\mathbf{Q}_2 \in \mathbb{R}^{p\times TK}$ to denote the two blocks of $\mathbf{Q}$ such that $\mathbf{Q}^\top = [\mathbf{Q}_1^\top, \mathbf{Q}_2^\top]$. By definition, we have

$$\mathbf{Q}^\top\mathbf{Q} = [\mathbf{Q}_1^\top, \mathbf{Q}_2^\top]\begin{bmatrix}\mathbf{Q}_1 \\ \mathbf{Q}_2\end{bmatrix} = \mathbf{Q}_1^\top\mathbf{Q}_1 + \mathbf{Q}_2^\top\mathbf{Q}_2 = \mathbf{I}_{TK}.$$

Note that the minimum singular value of $\mathbf{Q}_1 \in \mathbb{R}^{d\times TK}$ is zero since $d$ is a fixed number and $TK > d$. Therefore, it must hold that $\text{rank}(\mathbf{Q}_2) = TK$ and thus $\mathbf{Q}_2^\top\mathbf{Q}_2$ is positive definite. Let $\mathbf{r} = (r(\mathbf{x}_1),\dots,r(\mathbf{x}_i),\dots,r(\mathbf{x}_{TK}))^\top \in \mathbb{R}^{TK}$ denote the vector of all possible rewards. We further define $\mathbf{G} \in \mathbb{R}^{TKd\times p}$ and $\boldsymbol{\Phi} \in \mathbb{R}^{TKd}$ as follows

$$\mathbf{G} = \frac{1}{\sqrt{m}}\begin{bmatrix} \mathbf{g}(\mathbf{x}_1;\mathbf{w}^{(0)}) \\ \vdots \\ \mathbf{g}(\mathbf{x}_i;\mathbf{w}^{(0)}) \\ \vdots \\ \mathbf{g}(\mathbf{x}_{TK};\mathbf{w}^{(0)}) \end{bmatrix}, \quad \boldsymbol{\Phi} = \begin{bmatrix} \boldsymbol{\phi}(\mathbf{x}_{1,1};\mathbf{w}_0) \\ \vdots \\ \boldsymbol{\phi}(\mathbf{x}_{t,k};\mathbf{w}_{t-1}) \\ \vdots \\ \boldsymbol{\phi}(\mathbf{x}_{T,K};\mathbf{w}_{T-1}) \end{bmatrix}. \tag{D.3}$$

and $\boldsymbol{\Theta}, \boldsymbol{\Theta}_0 \in \mathbb{R}^{TK\times TKd}$ as follows

$$\boldsymbol{\Theta}^* = \begin{bmatrix} \boldsymbol{\theta}^{*\top} & & & & \\ & \ddots & & & \\ & & \boldsymbol{\theta}^{*\top} & & \\ & & & \ddots & \\ & & & & \boldsymbol{\theta}^{*\top} \end{bmatrix}, \quad \boldsymbol{\Theta}_0 = \begin{bmatrix} \boldsymbol{\theta}_0^\top & & & & \\ & \ddots & & & \\ & & \boldsymbol{\theta}_0^\top & & \\ & & & \ddots & \\ & & & & \boldsymbol{\theta}_0^\top \end{bmatrix}, \tag{D.4}$$

It can be verified that $\mathbf{\Psi} = \mathbf{PD}[\mathbf{Q}_1^\top, \mathbf{Q}_2^\top]$ and $\mathbf{PDQ}_2^\top = \mathbf{\Theta}_0\mathbf{G}$. Note that we have $\mathbf{Q}_2^\top\mathbf{Q}_2$ is positive definite by Assumption 4.3, which corresponds to the neural tangent kernel matrix defined on the first $L$ layers. Then we can define $\mathbf{w}^*$ as follows

$$\mathbf{w}^* = \mathbf{w}^{(0)} + 1/\sqrt{m}\mathbf{Q}_2(\mathbf{Q}_2^\top\mathbf{Q}_2)^{-1}\mathbf{D}^{-1}\mathbf{P}^\top(\mathbf{r} - \mathbf{\Theta}^*\mathbf{\Phi}). \tag{D.5}$$

We can verify that

$$\mathbf{\Theta}^*\mathbf{\Phi} + \sqrt{m}\mathbf{PDQ}_2^\top(\mathbf{w}^* - \mathbf{w}^{(0)}) = \mathbf{r}.$$

On the other hand, we have

$$\|\mathbf{w}^* - \mathbf{w}^{(0)}\|_2^2 \leq 1/m(\mathbf{r} - \mathbf{\Theta}^*\mathbf{\Phi})^\top\mathbf{PD}^{-1}(\mathbf{Q}_2^\top\mathbf{Q}_2)^{-1}\mathbf{D}^{-1}\mathbf{P}^\top(\mathbf{r} - \mathbf{\Theta}^*\mathbf{\Phi})$$
$$\leq 1/m(\mathbf{r} - \mathbf{\Theta}^*\mathbf{\Phi})^\top\mathbf{H}^{-1}(\mathbf{r} - \mathbf{\Theta}^*\mathbf{\Phi}),$$

which completes the proof. $\square$

### D.2 PROOF OF LEMMA C.2

Note that we can view the output of the last hidden layer $\phi(\mathbf{x}; \mathbf{w})$ defined in (2.4) as a vector-output neural network with weight parameter $\mathbf{w}$. The following lemma shows that the output of the neural network $\phi$ is bounded at the initialization.

**Lemma D.2** (Lemma 4.4 in Cao & Gu (2019b)). *Let $\delta \in (0, 1)$, and the width of the neural network satisfy $m \geq C_0 L \log(TKL/\delta)$. Then for all $t \in [T]$, $k \in [K]$ and $j \in [d]$, we have $|\phi_j(\mathbf{x}_{t,k}; \mathbf{w}^{(0)})| \leq C_1\sqrt{\log(TK/\delta)}$ with probability at least $1 - \delta$, where $\mathbf{w}^{(0)}$ is the initialization of the neural network.*

In addition, in a smaller neighborhood of the initialization, the gradient of the neural network $\phi$ is uniformly bounded.

**Lemma D.3** (Lemma B.3 in Cao & Gu (2019b)). *Let $\omega \leq C_0 L^{-6}(\log m)^{-3}$ and $\mathbf{w} \in \mathbb{B}(\mathbf{w}_0, \omega)$. Then for all $t \in [T]$, $k \in [K]$ and $j \in [d]$, the gradient of the neural network $\phi$ defined in (2.4) satisfies $\|\nabla_\mathbf{w}\phi_j(\mathbf{x}_{t,k}; \mathbf{w})\|_2 \leq C_1\sqrt{Lm}$ with probability at least $1 - TKL^2\exp(-C_2 m\omega^{2/3}L)$.*

The next lemma provides an upper bound on the gradient of the squared loss function defined in (3.3). Note that our definition of the loss function is slightly different from that in Allen-Zhu et al. (2019b) due to the output layer $\boldsymbol{\theta}_i$ and thus there is an additional term on the upper bound of $\|\boldsymbol{\theta}_i\|_2$ for all $i \in [T]$.

**Lemma D.4** (Theorem 3 in Allen-Zhu et al. (2019b)). *Let $\omega \leq C_0\delta^{3/2}/(T^{9/2}L^6\log^3 m)$. For all $\mathbf{w} \in \mathbb{B}(\mathbf{w}^{(0)}, \omega)$, with probability at least $1 - \exp(-C_1 m\omega^{2/3}L)$ over the randomness of $\mathbf{w}^{(0)}$, it holds that*

$$\|\nabla\mathcal{L}(\mathbf{w})\|_2^2 \leq \frac{C_2 Tm\mathcal{L}(\mathbf{w})\sup_{i=1,...,H}\|\boldsymbol{\theta}_i\|_2^2}{d}.$$

*Proof of Lemma C.2.* Fix the epoch number $q$ and we omit it in the subscripts in the rest of the proof when no confusion arises. Recall that $\mathbf{w}^{(s)}$ is the $s$-th iterate in Algorithm 2. Let $\delta > 0$ be any constant. Let $\omega$ be defined as follows.

$$\omega = \delta^{3/2}m^{-1/2}T^{-9/2}L^{-6}\log^{-3}(m). \tag{D.6}$$

We will prove by induction that with probability at least $1 - \delta$ the following statement holds for all $s = 0, 1, \ldots, n$

$$\phi_j(\mathbf{x}; \mathbf{w}^{(s)}) \leq C_0\sum_{h=0}^s \frac{\sqrt{\log(TK/\delta)}}{h+1}, \quad \text{for } \forall j \in [d]; \text{ and } \|\mathbf{w}_q^{(s)} - \mathbf{w}^{(0)}\| \leq \omega. \tag{D.7}$$

First note that (D.7) holds trivially when $s = 0$ due to Lemma D.2. Now we assume that (D.7) holds for all $j = 0, \ldots, s$. The loss function in (3.3) can be bounded as follows.

$$\mathcal{L}(\mathbf{w}^{(j)}) = \sum_{i=1}^{qH}(\boldsymbol{\theta}_i^\top\phi(\mathbf{x}_i; \mathbf{w}^{(j)}) - \widehat{r}_i)^2 \leq \sum_{i=1}^{qH} 2(\|\boldsymbol{\theta}_i\|_2^2 \cdot \|\phi(\mathbf{x}_i; \mathbf{w}^{(j)})\|_2^2 + 1).$$

By the update rule of $\boldsymbol{\theta}_t$, we have

$$\|\boldsymbol{\theta}_t\|_2 = \left\|\left(\lambda\mathbf{I} + \sum_{i=1}^{t}\boldsymbol{\phi}(\mathbf{x}_i;\mathbf{w}_{i-1})\boldsymbol{\phi}(\mathbf{x}_i;\mathbf{w}_{i-1})^\top\right)^{-1}\sum_{i=1}^{t}\boldsymbol{\phi}(\mathbf{x}_i;\mathbf{w}_{i-1})\widehat{\mathbf{r}}\right\|_2 \le 2d, \qquad \text{(D.8)}$$

where the inequality is due to Lemma C.5, which combined with (D.7) immediately implies

$$\mathcal{L}(\mathbf{w}^{(j)}) \le C_1 T d^3 \log(TK/\delta)\left(\sum_{h=0}^{j}\frac{1}{h+1}\right)^2 \le C_1 T d^3 \log(TK/\delta)\log^2 n. \qquad \text{(D.9)}$$

Substituting (D.8) and (D.9) into the inequality in Lemma D.4, we also have

$$\left\|\nabla\mathcal{L}(\mathbf{w}^{(j)})\right\|_2 \le C_2\sqrt{dTm\mathcal{L}(\mathbf{w}^{(j)})} \le C_3 d^2 T \log(n)\sqrt{m\log(TK/\delta)}. \qquad \text{(D.10)}$$

Now we consider $\mathbf{w}^{(s+1)}$. By triangle inequality we have

$$\begin{aligned}
\left\|\mathbf{w}^{(s+1)} - \mathbf{w}^{(0)}\right\|_2 &\le \sum_{j=0}^{s}\left\|\mathbf{w}^{(j+1)} - \mathbf{w}^{(j)}\right\|_2 \\
&= \sum_{j=0}^{s}\eta\left\|\nabla\mathcal{L}(\mathbf{w}^{(j)})\right\|_2 \\
&\le \sum_{j=0}^{s}\eta d^2 T\log(n)\sqrt{m\log(TK/\delta)}, \qquad \text{(D.11)}
\end{aligned}$$

where the last inequality is due to (D.10). If we choose the step size $\eta_q$ in the $q$-th epoch such that

$$\eta \le \frac{\omega}{d^2 T n \log(n)\sqrt{m\log(TK/\delta)}}, \qquad \text{(D.12)}$$

then we have $\|\mathbf{w}_q^{(s+1)} - \mathbf{w}^{(0)}\|_2 \le \omega$. Note that the choice of $m, \omega$ satisfies the condition in Lemma C.3. Thus we know $\phi_j(\mathbf{x};\mathbf{w})$ is almost linear in $\mathbf{w}$, which leads to

$$\begin{aligned}
|\phi_j(\mathbf{x};\mathbf{w}^{(s+1)})| &\le |\phi_j(\mathbf{x};\mathbf{w}^{(s)}) + \langle\nabla\phi_j(\mathbf{x};\mathbf{w}^{(s)}),\mathbf{w}^{(s+1)}-\mathbf{w}^{(s)}\rangle| + C_5\omega^{4/3}L^3 d^{-1/2}\sqrt{m\log m} \\
&\le \sum_{h=0}^{s}\frac{C\sqrt{\log(TK/\delta)}}{h+1} + \eta\sqrt{dm}\|\nabla\mathcal{L}(\mathbf{w}^{(s)})\|_2 + 2C_5\omega^{4/3}L^3 d^{-1/2}\sqrt{m\log m} \\
&\le \sum_{h=0}^{s}\frac{C_0\sqrt{\log(TK/\delta)}}{h+1} + C_3\eta\sqrt{dm}\sqrt{CT^2 d^4 m\log(TK/\delta)}\log n \\
&\quad + 2C_5\omega^{4/3}L^3 d^{-1/2}\sqrt{m\log m} \\
&= \sum_{h=0}^{s}\frac{C_0\sqrt{\log(TK/\delta)}}{h+1} + \frac{\omega\sqrt{dm}}{n} + 2C_5\omega^{4/3}L^3 d^{-1/2}\sqrt{m\log m}, \qquad \text{(D.13)}
\end{aligned}$$

where in the second inequality we used the induction hypothesis (D.7), Cauchy-Schwarz inequality and Lemma D.3, and the third inequality is due to (D.10). Note that the definition of $\omega$ in (D.6) ensures that $\omega\sqrt{dm} < 1/2$ and $\omega^{4/3}L^3 d^{-1/2}\sqrt{m\log m} \le m^{-1/6}T^{-6}L^{-5}d^{-1/2}\sqrt{\log m} \le 1/n$ as long as $m \ge n^6$. Plugging these two upper bounds back into (D.13) finishes the proof of (D.7).

Note that for any $t \in [T]$, we have $\mathbf{w}_t = \mathbf{w}_q^{(n)}$ for some $q = 1, 2, \ldots$. Since we have $\mathbf{w}_t \in \mathbb{B}(\mathbf{w},\omega)$, the gradient $\mathbf{g}(\mathbf{x};\mathbf{w}^{(0)})$ can be directly bounded by Lemma D.3, which implies $\|\mathbf{g}(\mathbf{x};\mathbf{w}^{(0)})\|_F \le C_6\sqrt{dLm}$. Applying (D.7) with $s = n$, we have the following bound of the neural network function $\phi(\mathbf{x};\mathbf{w}_q^{(n)}) = \phi(\mathbf{x};\mathbf{w}_t)$ for all $t$ in the $q$-th epoch

$$\|\boldsymbol{\phi}(\mathbf{x};\mathbf{w}_t)\|_2 \le C_0\sqrt{d\log(n)\log(TK/\delta)},$$

which completes the proof. In this proof, $\{C_j > 0\}_{j=0,\ldots,6}$ are constants independent of problem parameters. $\qquad\square$

### D.3 PROOF OF LEMMA C.4

The following lemma characterizes the concentration property of self-normalized martingales.

**Lemma D.5** (Theorem 1 in Abbasi-Yadkori et al. (2011)). *Let* $\{\xi\}_{t=1}^{\infty}$ *be a real-valued stochastic process and* $\{\mathbf{x}_t\}_{t=1}^{\infty}$ *be a stochastic process in* $\mathbb{R}^d$. *Let* $\mathcal{F}_t = \sigma(\mathbf{x}_1, \ldots, \mathbf{x}_{t+1}, \xi - 1, \ldots, \xi_t)$ *be a* $\sigma$*-algebra such that* $\mathbf{x}_t$ *and* $\xi_t$ *are* $\mathcal{F}_{t-1}$*-measurable. Let* $\mathbf{A}_t = \lambda \mathbf{I} + \sum_{s=1}^{t} \mathbf{x}_s \mathbf{x}_s^{\top}$ *for some constant* $\lambda > 0$ *and* $S_t = \sum_{s=1}^{t} \xi_s \mathbf{x}_i$. *If we assume* $\xi_t$ *is* $\nu$*-subGaussian conditional on* $\mathcal{F}_{t-1}$, *then for any* $\eta \in (0, 1)$, *with probability at least* $1 - \delta$, *we have*

$$\|S_t\|_{\mathbf{A}_t^{-1}}^2 \leq 2\nu^2 \log\left(\frac{\det(\mathbf{A}_t)^{1/2} \det(\lambda \mathbf{I})^{-1/2}}{\delta}\right).$$

*Proof of Lemma C.4.* Let $\mathbf{\Phi}_t = [\phi(\mathbf{x}_{1,a_1}; \mathbf{w}_0), \ldots, \phi(\mathbf{x}_{t,a_t}; \mathbf{w}_{t-1})] \in \mathbb{R}^{d \times t}$ be the collection of feature vectors of the chosen arms up to time $t$ and $\widehat{\mathbf{r}}_t = (\widehat{r}_1, \ldots, \widehat{r}_t)^{\top}$ be the concatenation of all received rewards. According to Algorithm 1, we have $\mathbf{A}_t = \lambda \mathbf{I} + \mathbf{\Phi}_t \mathbf{\Phi}_t^{\top}$ and thus

$$\boldsymbol{\theta}_t = \mathbf{A}_t^{-1} \mathbf{b}_t = (\lambda \mathbf{I} + \mathbf{\Phi}_t \mathbf{\Phi}_t^{\top})^{-1} \mathbf{\Phi}_t \widehat{\mathbf{r}}_t.$$

By Lemma C.1, the underlying reward generating function $r_t = r(\mathbf{x}_{t,a_t}) = \mathbb{E}[\widehat{r}(\mathbf{x}_{t,a_t}) | \mathbf{x}_{t,a_t}]$ can be rewritten as

$$r_t = \langle \boldsymbol{\theta}^*, \phi(\mathbf{x}_{t,a_t}; \mathbf{w}_{t-1}) \rangle + \boldsymbol{\theta}_0^{\top} \mathbf{g}(\mathbf{x}_{t,a_t}; \mathbf{w}^{(0)})(\mathbf{w}^* - \mathbf{w}^{(0)}).$$

By the definition of the reward in (2.1) we have $\widehat{r}_t = r_t + \xi_t$. Therefore, it holds that

$$\boldsymbol{\theta}_t = \mathbf{A}_t^{-1} \mathbf{\Phi}_t \mathbf{\Phi}_t^{\top} \boldsymbol{\theta}^* + \mathbf{A}_t^{-1} \sum_{s=1}^{t} \phi(\mathbf{x}_{s,a_s}; \mathbf{w}_{s-1})(\boldsymbol{\theta}_0^{\top} \mathbf{g}(\mathbf{x}_{s,a_s}; \mathbf{w}^{(0)})(\mathbf{w}^* - \mathbf{w}^{(0)}) + \xi_s)$$

$$= \boldsymbol{\theta}^* - \lambda \mathbf{A}_t^{-1} \boldsymbol{\theta}^* + \mathbf{A}_t^{-1} \sum_{s=1}^{t} \phi(\mathbf{x}_{s,a_s}; \mathbf{w}_{s-1})(\boldsymbol{\theta}_0^{\top} \mathbf{g}(\mathbf{x}_{s,a_s}; \mathbf{w}^{(0)})(\mathbf{w}^* - \mathbf{w}^{(0)}) + \xi_s).$$

Note that $\mathbf{A}_t$ is positive definite as long as $\lambda > 0$. Therefore $\|\cdot\|_{\mathbf{A}_t}$ and $\|\cdot\|_{\mathbf{A}_t}$ are well defined norms. Then for any $\delta \in (0, 1)$ by triangle inequality we have

$$\|\boldsymbol{\theta}_t - \boldsymbol{\theta}^* - \mathbf{A}_t^{-1} \mathbf{\Phi}_t \mathbf{\Theta}_t \mathbf{G}_t(\mathbf{w}^* - \mathbf{w}^{(0)})\|_{\mathbf{A}_t} \leq \lambda \|\boldsymbol{\theta}^*\|_{\mathbf{A}_t^{-1}} + \|\mathbf{\Phi}_t \boldsymbol{\xi}_t\|_{\mathbf{A}_t^{-1}}$$

$$\leq \nu \sqrt{2 \log\left(\frac{\det(\mathbf{A}_t)^{1/2} \det(\lambda \mathbf{I})^{-1/2}}{\delta}\right)} + \lambda^{1/2} M$$

holds with probability at least $1 - \delta$, where in the last inequality we used Lemma D.5 and the fact that $\|\boldsymbol{\theta}^*\|_{\mathbf{A}_t^{-1}} \leq \lambda^{-1/2} \|\boldsymbol{\theta}^*\|_2 \leq \lambda^{-1/2} M$ by Lemma C.1. Plugging the definition of $\mathbf{\Phi}_t$, $\mathbf{\Theta}_t$ and $\mathbf{G}_t$ and apply Lemma C.6, we further have

$$\left\|\boldsymbol{\theta}_t - \boldsymbol{\theta}^* - \mathbf{A}_t^{-1} \sum_{s=1}^{t} \phi(\mathbf{x}_{s,a_s}; \mathbf{w}_{s-1}) \boldsymbol{\theta}_0^{\top} \mathbf{g}(\mathbf{x}_{s,a_s}; \mathbf{w}^{(0)})(\mathbf{w}^* - \mathbf{w}^{(0)})\right\|_{\mathbf{A}_t}$$

$$\leq \nu \sqrt{2\big(d \log(1 + t(\log HK)/\lambda) + \log 1/\delta\big)} + \lambda^{1/2} M,$$

where we used the fact that $\|\phi(\mathbf{x}; \mathbf{w})\|_2 \leq C\sqrt{d \log HK}$ by Lemma C.2. $\qquad\square$

### D.4 PROOF OF LEMMA C.5

We now prove the technical lemma that upper bounds $\|\mathbf{A}_t^{-1} \sum_{s=1}^{t} \phi_s \zeta_s\|_2$.

*Proof of Lemma C.5.* We first construct auxiliary vectors $\widetilde{\phi}_t \in \mathbb{R}^{d+1}$ and matrices $\mathbf{B}_t \in \mathbb{R}^{(d+1) \times (d+1)}$ for all $t = 1, \ldots$ in the following way:

$$\widetilde{\phi}_t = \begin{bmatrix} G^{-1} \phi_t \\ \sqrt{1 - G^{-2} \|\phi_t\|_2^2} \end{bmatrix}, \quad \mathbf{B}_t = \begin{bmatrix} \mathbf{A}_t^{-1} & \mathbf{0}_d \\ \mathbf{0}_d^{\top} & 0 \end{bmatrix}, \tag{D.14}$$

where $\mathbf{0}_d \in \mathbb{R}^d$ is an all-zero vector. Then by definition we immediately have

$$\left\| \mathbf{A}_t^{-1} \sum_{s=1}^t \boldsymbol{\phi}_s \zeta_s \right\|_2 = \left\| \mathbf{B}_t \sum_{s=1}^t \widetilde{\boldsymbol{\phi}}_s \zeta_s \right\|_2. \tag{D.15}$$

For all $s = 1, 2 \ldots$, let $\{\beta_{s,j}\}_{j=1}^{d+1}$ be the coefficients of the decomposition of $U^{-1}\zeta_s \widetilde{\boldsymbol{\phi}}_s$ on the natural basis. Specifically, let $\{\mathbf{e}_1, \ldots, \mathbf{e}_{d+1}\}$ be the natural basis of $\mathbb{R}^{d+1}$ such that the entries of $\mathbf{e}_j$ are all zero except the $j$-th entry which equals 1. Then we have

$$U^{-1}\zeta_s \widetilde{\boldsymbol{\phi}}_s = \sum_{j=1}^d \beta_{s,j}\mathbf{e}_j, \quad \forall s = 1, 2, \ldots \tag{D.16}$$

We can conclude that $|\beta_{s,j}| \leq 1$ since $|\zeta_s| \leq U$ and $\|\widetilde{\boldsymbol{\phi}}_s\|_2 \leq 1$. Moreover, it is easy to verify that $\|\widetilde{\boldsymbol{\phi}}_t\|_2 = 1$ for all $t \geq 1$. Therefore, we have

$$\left\| \mathbf{B}_t \sum_{s=1}^t \widetilde{\boldsymbol{\phi}}_s \zeta_s \right\|_2 = \left\| \mathbf{B}_t \sum_{s=1}^t \widetilde{\boldsymbol{\phi}}_s \widetilde{\boldsymbol{\phi}}_s^\top \widetilde{\boldsymbol{\phi}}_s \zeta_s \right\|_2$$

$$= \left\| \mathbf{B}_t \sum_{s=1}^t \widetilde{\boldsymbol{\phi}}_s \widetilde{\boldsymbol{\phi}}_s^\top U \sum_{j=1}^d \beta_{s,j}\mathbf{e}_j \right\|_2$$

$$= U \left\| \sum_{j=1}^d \mathbf{B}_t \sum_{s=1}^t \widetilde{\boldsymbol{\phi}}_s \widetilde{\boldsymbol{\phi}}_s^\top \beta_{s,j}\mathbf{e}_j \right\|_2$$

$$\leq U \sum_{j=1}^d \left\| \mathbf{B}_t \sum_{s=1}^t \widetilde{\boldsymbol{\phi}}_s \widetilde{\boldsymbol{\phi}}_s^\top \beta_{s,j} \right\|_2$$

$$= U \sum_{j=1}^d \left\| \mathbf{A}_t^{-1} \sum_{s=1}^t \boldsymbol{\phi}_s \boldsymbol{\phi}_s^\top \beta_{s,j} \right\|_2, \tag{D.17}$$

where the inequality is due to triangle inequality and the last equation is due to the definition of $\widetilde{\boldsymbol{\phi}}_t$ and $\mathbf{B}_t$ in (D.14). For each $j = 1, \ldots, d+1$, we have

$$\left\| \mathbf{A}_t^{-1} \sum_{s=1}^t \boldsymbol{\phi}_s \boldsymbol{\phi}_s^\top \beta_{s,j} \right\|_2 = \left\| \mathbf{A}_t^{-1} \sum_{s\in[t]:\beta_{s,j}\geq 0} \boldsymbol{\phi}_s \boldsymbol{\phi}_s^\top \beta_{s,j} + \mathbf{A}_t^{-1} \sum_{s\in[t]:\beta_{s,j}<0} \boldsymbol{\phi}_s \boldsymbol{\phi}_s^\top \beta_{s,j} \right\|_2$$

$$\leq \left\| \mathbf{A}_t^{-1} \sum_{s\in[t]:\beta_{s,j}\geq 0} \boldsymbol{\phi}_s \boldsymbol{\phi}_s^\top \beta_{s,j} \right\|_2 + \left\| \mathbf{A}_t^{-1} \sum_{s\in[t]:\beta_{s,j}<0} \boldsymbol{\phi}_s \boldsymbol{\phi}_s^\top (-\beta_{s,j}) \right\|_2. \tag{D.18}$$

Since we have $|\beta_{s,j}| \leq 1$, it immediately implies

$$\mathbf{A}_t = \lambda \mathbf{I} + \sum_{s=1}^t \boldsymbol{\phi}_s \boldsymbol{\phi}_s^\top \succ \sum_{s\in[t]:\beta_{s,j}\geq 0} \boldsymbol{\phi}_s \boldsymbol{\phi}_s^\top \beta_{s,j},$$

$$\mathbf{A}_t = \lambda \mathbf{I} + \sum_{s=1}^t \boldsymbol{\phi}_s \boldsymbol{\phi}_s^\top \succ \sum_{s\in[t]:\beta_{s,j}<0} \boldsymbol{\phi}_s \boldsymbol{\phi}_s^\top (-\beta_{s,j}).$$

Further by the fact that $\|\mathbf{A}^{-1}\mathbf{B}\|_2 \leq 1$ for any $\mathbf{A} \succ \mathbf{B} \succeq 0$, combining the above results with (D.18) yields

$$\left\| \mathbf{A}_t^{-1} \sum_{s=1}^t \boldsymbol{\phi}_s \boldsymbol{\phi}_s^\top \beta_{s,j} \right\|_2 \leq 2.$$

Finally, substituting the above results into (D.17) and (D.15) we have

$$\left\| \mathbf{A}_t^{-1} \sum_{s=1}^t \boldsymbol{\phi}_s \zeta_s \right\|_2 \leq 2Ud,$$

which completes the proof. $\qquad\square$

