# OpenReview forum: "Neural Contextual Bandits with Deep Representation and Shallow Exploration"
_ICLR.cc/2022/Conference — ICLR 2022 Poster_

### Official Review · Reviewer_hExw · 2021-11-01

**Correctness:** 4
**Technical Novelty And Significance:** 3
**Empirical Novelty And Significance:** 3
**Recommendation:** 8
**Confidence:** 4

**Main Review:**

The paper is well written. The studied problem is significant. The theoretical results are interesting.

The reviewer has some comments and questions.

If Neural-LinUCB reduces the computational cost of NeuralUCB, it remains very high in comparison to LinUCB due to the episodic retraining of the deep neural network. Worst, the computational cost over-linearly increases with T since the deep neural network is retrained every H episode with all data. Neural-LinUCB cannot be used if T is too large.
Do the authors have any leads for handling this practical problem?

The theoretical results are very interesting since it concerns the best performer in contextual bandits, but they are still quite weak.

First, the regret upper bound depends on r-\tilde(r), the regret between the obtained mapping of the reward function and the best mapping. It seems not oblivious that the right term of R_T in corollary 4.6 does not scale in \sqrt(T).

Second, the condition m \geq T^3 seems strong and confirms that the proposed algorithm does not scale with T.

AFTER REBUTTAL
I thank the authors for their answers. The analytical results on the contextual bandits with Deep Neural Networks are not totally news, as well as the algorithm with shallow exploration on the last layer, but together, they constitute a significant step. The proposed algorithm is a LinUCB-like algorithm, but that uses the powerful representation obtained in the last layer of a Deep Neural Nertwork to perform LinUCB. As a result, a clear improvement of performances in comparison to LinUCB, and moreover this fixes a known issue of LinUCB: due to the matrix inversion, LinUCB cannot process more than few hundreds of features. Here due to the use of a representation layer, this issue is fixed. An efficient Neural Contextual Bandit algorithm is analyzed, and this could have a deep impact on the bandit community. That is why I vote for acceptance.


**Summary Of The Paper:**

This paper study a novel contextual bandit algorithm: Neural-LinUCB.
As in (Riquelme et al 2019), the idea of this algorithm is based on decoupling deep representation learning and exploration. A deep neural network learns the mapping between the context $x_{t,a_t}$, while a linear bandit, OFUL (Abbasi-Yadkori 2011), chooses the arm to play.
In contrast to (Riquelme et al 2019), a regret upper bound of the algorithm a regret upper bound of the algorithm is stated in Corollary 4.6. The proposed algorithm is also an improvement over NeuralUCB (Zhou et al 2020) for two reasons: the computational cost of the exploration is lesser, since the exploration is only done in the last layer of weights, and the regret upper bound is tighter. Indeed, in contrast to (Zhou et al 2020) it does not depend on the dimension of the tangent kernel matrix, which can be in O(KT).
Experiments, done on four contextual bandit problems, show that Neura-lLinUCB outperforms LinUCB and performs as well as NeuralUCB and NeuralTS.


**Summary Of The Review:**

The paper is well written. The studied problem is significant. The theoretical results are interesting. The reviewer accepts this paper.

---

> ### Author Response · Authors · 2021-11-18
> **Response to Reviewer hExw**
>
> Thank you for your helpful comments. We address your comments point by point as follows.
>
> ---
> **Q**: “If Neural-LinUCB reduces the computational cost of NeuralUCB, it remains very high in comparison to LinUCB due to the episodic retraining of the deep neural network. Worst, the computational cost over-linearly increases with T since the deep neural network is retrained every H episode with all data. Neural-LinUCB cannot be used if T is too large. Do the authors have any leads for handling this practical problem?”
> **A**: Similar to Zhou et al. (2020), we use gradient descent (GD) in theoretical analysis to simplify the analysis. It can be replaced by stochastic gradient descent (SGD) and it only affects the analysis on the representation power of the neural network, which can be addressed by the more involved NTK analysis of SGD in Allen-Zhu et al., (2019) and Zou et al., (2019). In practice, we use SGD rather than GD to update the weight parameters. We will add the details of the training procedure in our revision. On the other hand, we can modify our algorithm in a way similar to the rarely switching algorithm proposed in Abbasi-yadkori et al. 2011. Instead of retraining the deep neural networks in each round, the modified algorithm will only retrain the neural network when there is a sufficient change to the collected data. This rarely switching algorithm will reduce the number of retraining from $T$ to $\log T$. We will study it in our future work.
>
> ---
> **Q**: “The regret upper bound depends on r-\tilde(r), the regret between the obtained mapping of the reward function and the best mapping. It seems not oblivious that the right term of R_T in corollary 4.6 does not scale in \sqrt(T).”
> **A**: Note that the term that depends on $r-\tilde{r}$ in our regret has $m^{1/6}$ in its denominator. Therefore, we can always choose $m$ to be sufficiently large, such that this term scales in \sqrt{T}.
>
> ---
> **Q**: “The condition m \geq T^3 seems strong and confirms that the proposed algorithm does not scale with T.”
> **A**: We would also like to point out that the theoretical value of $m$ is a sufficient condition to prove our results. However, it is not necessary in practice based on our ablation study in Appendix A.2. In other words, our theory is conservative since the current NTK-based analysis of deep learning is still very limited, which requires extra-wide neural network width. Nevertheless, the main goal of our work is to show that the idea of deep representation learning and shallow exploration works and can achieve a sublinear regret, rather than improving the NTK-based deep learning theory. So we believe our work is a good addition to the existing literature on neural bandit algorithms, and there is still room for improvement in its theory.

---

> > ### Comment · Reviewer_hExw · 2021-11-22
> > **After rebuttal**
> >
> > I thank the authors for their answers.
> > The analytical results on the contextual bandits with Deep Neural Networks are not totally news, as well as the algorithm with shallow exploration on the last layer, but together, they constitute a significant step. The proposed algorithm is a LinUCB-like algorithm, but that uses the powerful representation obtained in the last layer of a Deep Neural Nertwork to perform LinUCB. As a result, a clear improvement of performances in comparison to LinUCB, and moreover this fixes a known issue of LinUCB: due to the matrix inversion, LinUCB cannot process more than few hundreds of features. Here due to the use of a representation layer, this issue is fixed.
> > An efficient Neural Contextual Bandit algorithm is analyzed, and this could have a deep impact on the bandit community.
> > That is why I vote for acceptance.

---

> > > ### Author Response · Authors · 2021-11-22
> > > **Thank you for your reply**
> > >
> > > Thank you for your reply and your supportive comments. We appreciate it very much!

---

### Official Review · Reviewer_hPGC · 2021-11-02

**Correctness:** 4
**Technical Novelty And Significance:** 2
**Empirical Novelty And Significance:** 2
**Recommendation:** 3
**Confidence:** 4

**Main Review:**

Strengths:

1. The studied neural contextual bandit is very interesting and important in the area of applying neural networks to online learning.
2. The proposed algorithm significantly reduces the computation costs when compared to existing neural contextual bandit algorithms.

Weakness:

While the proposed idea, i.e., transforming the raw feature vector using the last hidden layer of a deep ReLU neural network and adopting an UCB approach to explore in the last linear layer, is simple and effective, this idea is inspired by existing Thompson Sampling based work [Riquelme et al. 2018]. In addition, several techniques in the regret analysis of this paper are similar to existing works, e.g., [Riquelme et al. 2018; Zhou et al. 2020], which reduce the technical novelty of this paper.


---After Rebuttal---

I thank the authors for their response. However, my concern on limited contribution and novelty was not well addressed. See my following comments for details. I plan to stick my score 3, i.e., reject.

1. Algorithmic design.

The high-level idea of decoupling representation learning and exploration is originated from Riquelme et al. (2018).
As we know, Thompson Sampling and UCB are two standard algorithmic styles in bandits.
While Riquelme et al. (2018) is a Thompson Sampling based algorithm and you design a UCB based algorithm, I do not think that adapting an existing idea in Thompson Sampling type algorithm to the UCB type algorithm is a significant contribution in algorithmic design.

2. Theoretical analysis

I appreciate that you propose the first theoretical analysis for the idea of decoupling representation learning and exploration on your UCB type algorithm, while prior work [Riquelme et al. (2018)] does not give theoretical analysis on their Thompson Sampling type algorithm.

However, it is well known that the Thompson Sampling based analysis is intrinsically more complicated and harder than the UCB based analysis, which reduces the technical challenges of analyzing this idea on your UCB type algorithm to some degree.

I go through the new technical contributions you mentioned, i.e., Lemma C.1 and Lemma C.4. The proofs of Lemma C.1 and Lemma C.4 also rely on prior NTK works, e.g., Theorem 3.1 in Arora et al. (2019b), Lemma 4.4 in Cao & Gu (2019b), Lemma B.3 in Cao & Gu (2019b), Theorem 3 in Allen-Zhu et al. (2019b).
In my opinon, the analysis is a standard combination of existing techniques from prior neural contextual bandit [Zhou et al. 2020] and NTK [Arora et al. (2019b); Cao & Gu (2019b); Allen-Zhu et al. (2019b)] papers, following the high-level idea from [Riquelme et al. (2018)].

From my view, the biggest technical contribution of this paper is that you borrow many techniques from neural contextual linear bandit [Zhou et al. 2020] and NTK [Arora et al. (2019b); Cao & Gu (2019b); Allen-Zhu et al. (2019b)] to execute the high-level idea of decoupling representation learning and exploration from [Riquelme et al. (2018)] in theoretical analysis. However, I think that the contribution and novelty is weak.

----Additional Response----

I have read the authors' further response. Unfortunately, my concern on technical novelty and contributions was not relieved.

This paper borrows the high-level idea of decoupling representation learning and exploration from [Riquelme et al. (2018)] to design a UCB-type algorithm, and analyzes it by borrowing many techniques from existing UCB-type neural contextural bandit/NTK papers [Zhou et al. 2020;Arora et al. (2019b); Cao & Gu (2019b); Allen-Zhu et al. (2019b)].
Although this paper well executes the high-level idea in [Riquelme et al. (2018)] and completes the analysis (in the more well-studied UCB style), I do not think this is a big contribution.

The technical novelty/contribution of this paper is very limited.
Not only the theoretical analysis of this paper is mostly built upon existing works  [Zhou et al. 2020;Arora et al. (2019b); Cao & Gu (2019b); Allen-Zhu et al. (2019b)], but also the idea of algorithm design also comes from existing work  [Riquelme et al. (2018)].

My opinion can be supported by the following facts in the proof details in this paper:

In Appendix C, to prove the main theorem, the authors wrote Lemmas C.1-C.6, among which Lemmas C.1, C.3, C.6 (or their similar forms) have already appeared in existing papers [Zhou et al. 2020], [Cao & Gu 2019b], [Abbasi-Yadkori et al. 2011], respectively.

In Appendix D, in the proof of (the only new) Lemmas C.2, C.4, C.5, the authors further borrowed many existing techniques, including Lemmas D.1 [Arora et al. (2019b)], D.2 [Cao & Gu 2019b], D.3 [Cao & Gu 2019b], D.4 [Allen-Zhu et al. (2019b)], D.5 [Abbasi-Yadkori et al. 2011].
In addition, the authors also followed the well-known analysis procedure of self-normalized concentration bound (in the proof of Lemma D.5) and confidence ellipsoid  (in the proof of Lemma D.4) from [Abbasi-Yadkori et al. 2011].

In conclusion, unfortunately, I do not think this paper has sufficient technical novelty. I will stick to my recommendation 'reject'.



**Summary Of The Paper:**

This paper studies neural contextual bandits and proposes an algorithm that transforms the raw feature vector using the last hidden layer of a deep ReLU neural network, and uses an UCB approach to explore in the last linear layer. Compared with existing neural contextual bandit algorithms, the proposed algorithm attains computation efficiency. Regret guarantees and empirical results are provided to demonstrate the effectiveness of the proposed algorithms

**Summary Of The Review:**

Overall, I think that the studied neural contextual bandit problem in this paper is very important. The idea behind the proposed algorithm is simple and effective. Theoretical and experimental results are provided to demonstrate the effectiveness of the proposed algorithm. However, this paper borrows the idea and analytical techniques in prior works and lacks theoretical novelty.

---

> ### Author Response · Authors · 2021-11-18
> **Response to Reviewer hPGC**
>
> Thank you for your helpful comments. We address your comment on the technical novelty of our paper as follows.
>
> ---
> **Q**: “While the proposed idea, i.e., transforming the raw feature vector using the last hidden layer of a deep ReLU neural network and adopting a UCB approach to explore in the last linear layer, is simple and effective, this idea is inspired by existing Thompson Sampling based work [Riquelme et al. 2018]. In addition, several techniques in the regret analysis of this paper are similar to existing works, e.g., [Riquelme et al. 2018; Zhou et al. 2020], which reduce the technical novelty of this paper.”
> **A**: First of all, we want to point out that our algorithm design is also a novel contribution besides our theoretical analysis. The Neural-Linear algorithm provided in Riquelme et al. (2018) is only based on a generic form of Thompson sampling. There are no special algorithm designs nor any provable guarantees for the Neural-Linear algorithm. In contrast, our algorithm Neural-LinUCB is carefully designed such that we for the first time show when and how the high-level idea of decoupling representation learning and exploration works in both theory and practice.
>
> Regarding the regret analysis, we would like to reiterate that there is no regret analysis in Riquelme et al. (2018) at all. Compared with Zhou et al. (2020), which treats the weight of the neural network as a whole, Neural-LinUCB in our paper decouples the deep representation learning from the UCB exploration. This decoupling imposes more challenges in the analysis since we need to separately show that (1) the network parameter up to the last hidden layer converges, and (2) the last layer parameter is in a confidence ball of the true parameter. No existing theoretical analysis can deal with the setting in our paper since all of them rely on the full exploration of all weight parameters. In particular, our regret analysis relies on Lemma C.1 and Lemma C.4 to show that the deep representation part and the shallow exploration part can be separated, which are new and not present in any existing work on neural bandits including Zhou et al, (2020).

---

> > ### Comment · Reviewer_hPGC · 2021-11-21
> > **Comments after the Authors' Response**
> >
> > Thank you for your reply.
> >
> > 1. Algorithmic design.
> >
> > The high-level idea of decoupling representation learning and exploration is originated from Riquelme et al. (2018).
> > As we know, Thompson Sampling and UCB are two standard algorithmic styles in bandits.
> > While Riquelme et al. (2018) is a Thompson Sampling based algorithm and you design a UCB based algorithm, I do not think that adapting an existing idea in Thompson Sampling type algorithm to the UCB type algorithm is a significant contribution in algorithmic design.
> >
> > 2. Theoretical analysis
> >
> > I appreciate that you propose the first theoretical analysis for the idea of decoupling representation learning and exploration on your UCB type algorithm, while prior work [Riquelme et al. (2018)] does not give theoretical analysis on their Thompson Sampling type algorithm.
> >
> > However, it is well known that the Thompson Sampling based analysis is intrinsically more complicated and harder than the UCB based analysis, which reduces the technical challenges of analyzing this idea on your UCB type algorithm to some degree.
> >
> > I go through the new technical contributions you mentioned, i.e., Lemma C.1 and Lemma C.4. The proofs of Lemma C.1 and Lemma C.4 also rely on prior NTK works, e.g., Theorem 3.1 in Arora et al. (2019b), Lemma 4.4 in Cao & Gu (2019b), Lemma B.3 in Cao & Gu (2019b), Theorem 3 in Allen-Zhu et al. (2019b).
> > In my opinon, the analysis is a standard combination of existing techniques from prior neural contextual bandit [Zhou et al. 2020] and NTK [Arora et al. (2019b); Cao & Gu (2019b); Allen-Zhu et al. (2019b)] papers, following the high-level idea from [Riquelme et al. (2018)].
> >
> > From my view, the biggest technical contribution of this paper is that you borrow many techniques from neural contextual linear bandit [Zhou et al. 2020] and NTK [Arora et al. (2019b); Cao & Gu (2019b); Allen-Zhu et al. (2019b)] to execute the high-level idea of decoupling representation learning and exploration from [Riquelme et al. (2018)] in theoretical analysis. However, I think that the contribution and novelty is weak.

---

> > > ### Author Response · Authors · 2021-11-21
> > > **Re: Comments after the Authors' Response (Part I)**
> > >
> > > Thank you for your reply. We respectfully disagree with you on the technical novelty and contribution of our paper.
> > >
> > > First of all, we admit that our work is built upon prior work on the analysis of the neural tangent kernel and neural contextual bandits, and the experimental design in decoupling representation learning and exploration. However, our provably efficient algorithm is by no means a trivial combination of these components. We don’t think it is fair to judge our contribution purely based on whether we rely on existing work to conduct our research. In particular, we address your comments as follows from the perspectives of both algorithm design and theoretical analysis.
> > >
> > >
> > > ## From the perspective of algorithm design:
> > > ---
> > > **Q**: “I do not think that adapting an existing idea in Thompson Sampling type algorithm to the UCB type algorithm is a significant contribution”
> > > **A**: We would like to argue that even some details about the algorithm design such as the inflation parameter in the variance of Thompson sampling can yield a very different regret bound or exploding regret. See Hamidi, Nima, and Mohsen Bayati. "On worst-case regret of linear Thompson sampling." arXiv preprint arXiv:2006.06790 (2020) for an example. In [Riquelme et al. (2018)], there is only a high-level idea of decoupling the representation learning and the exploration in Thompson sampling. However, there are several key components missing in their paper that guarantee the performance of the algorithm: how to sample from the posterior distribution, how to choose the parameter for the variance, how often to update the neural network for the representation learning, what algorithm to use in the updating of the representation, etc. In contrast, in our Neural-LinUCB algorithm, we provide every single detail of the algorithm design as well as a rigorous analysis, which is far beyond the simple high-level idea of decoupling representation learning and exploration. To sum up, except that our algorithm and NeuralLinear share the same high-level idea, they are drastically different algorithms. We don’t think it’s fair to criticize our algorithm based on a related work that only shares the high-level idea, especially given that we have carefully discussed and compared our algorithm with this work.

---

> > > > ### Comment · Reviewer_hPGC · 2021-11-26
> > > > **Response to the authors' rebuttal**
> > > >
> > > > Thank you for your reply.
> > > >
> > > > 1. What I meant for Thompson Sampling based analysis/[Riquelme et al. (2018)]
> > > >
> > > > I did not mean to compare your theoretical analysis with Thompson Sampling based analysis. I meant that the Thompson Sampling-type algorithm [Riquelme et al. (2018)] does not provide regret analysis is understandable, because analyzing Thompson Sampling  type algorithm is very difficult and there is no such analysis in the neural contextual bandit literature.
> > > >
> > > > As for your paper, you borrow the high-level idea of decoupling representation learning and exploration from [Riquelme et al. (2018)] to design a UCB-type algorithm, and analyze it by borrowing many techniques from existing UCB-type neural contextural bandit/NTK papers [Zhou et al. 2020;Arora et al. (2019b); Cao & Gu (2019b); Allen-Zhu et al. (2019b)].
> > > > Algthough I understand that you well executed the high-level idea in [Riquelme et al. (2018)] and completed the analysis (in the more well-studied UCB style), I do not think this is a big contribution.
> > > >
> > > > 2. Technical novelty of this paper/My opinions on borrowing techniques from existing works
> > > >
> > > > I did not mean that a good research work should not borrow any technique from existing papers. However, the analysis of this paper is mostly built upon existing works  [Zhou et al. 2020;Arora et al. (2019b); Cao & Gu (2019b); Allen-Zhu et al. (2019b)]. If you just borrow existing tools to prove  a new idea/insight or solve a new problem model, I think it is OK and you provide original ideas/insights. However, the idea of your algorithm design also comes from existing work  [Riquelme et al. (2018)], and you executed this idea by borrowing existing techniques (and probably adding some new analysis). I do not think this work has sufficient novelty.
> > > >
> > > > 3. Details of your analysis in Appendix
> > > >
> > > > In Appendix C, to prove the main theorem, you wrote Lemmas C.1-C.6, among which Lemmas C.1, C.3, C.6 or their similar forms have appeared in existing papers [Zhou et al. 2020], [Cao & Gu 2019b], [Abbasi-Yadkori et al. 2011], respectively.
> > > >
> > > > In Appendix D, in the proof of (remaining new) Lemmas C.2, C.4, C.5, you further borrowed four existing techniques, including Lemmas D.1 [Arora et al. (2019b)], D.2 [Cao & Gu 2019b], D.3 [Cao & Gu 2019b], D.4 [Allen-Zhu et al. (2019b)], D.5 [Abbasi-Yadkori et al. 2011].
> > > > In addition, you also followed the well-known analysis procedure of self-normalized concentration bound (in the proof of Lemma D.5) and confidence ellipsoid  (in the proof of Lemma D.4) from [Abbasi-Yadkori et al. 2011].
> > > >
> > > > Due to the above reasons, unfortunately, I do not think this paper has sufficient technical novelty.

---

> > > > > ### Author Response · Authors · 2021-11-28
> > > > > **Re: Response to the authors' rebuttal**
> > > > >
> > > > > Thank you for your reply. It seems that you are repeating your previous comments, which have been answered by our previous response. We are sorry that you still do not recognize the novelty of our paper. We will emphasize the contributions of our paper again:
> > > > >
> > > > > - We propose a new UCB-type contextual bandit algorithm based on the idea of decoupling the deep representation and the exploration in neural network-based bandit problems. Our algorithm is much more computationally efficient than existing neural contextual bandit algorithms.
> > > > >
> > > > > - We prove a sublinear regret for our proposed algorithm, which matches the results in existing works. Our work is also the first to provably show that decoupling the deep representation and exploration works.
> > > > >
> > > > > - We show that the proposed algorithm achieves comparable or better empirical performance in terms of regret and saves a lot of computation cost compared with other neural contextual bandit algorithms.

---

> > > ### Author Response · Authors · 2021-11-21
> > > **Re: Comments after the Authors' Response  (Part II)**
> > >
> > > ## From the perspective of theoretical analysis:
> > > ---
> > > **Q**: “It is well known that the Thompson Sampling based analysis is intrinsically more complicated and harder than the UCB based analysis, which reduces the technical challenges of analyzing this idea on your UCB type algorithm to some degree”
> > > **A**: We don’t understand the reviewer’s point of comparing our analysis with Thomson sampling-based analysis, which does not exist in the literature yet for the setting we are studying. Note that [Riquelme et al. (2018)] is a purely empirical study that provides a benchmark comparison for many existing heuristic algorithms, which is a paper with a totally different focus from ours. Also, we think the value of an algorithm should not be judged mainly based on how complicated and hard the analysis is. Given that our algorithm achieves comparable or even better results than NeuralLinear in our experiments, why do we want to pursue the Thompson sampling-based analysis?
> > >
> > > ---
> > > **Q**: “The biggest technical contribution of this paper is that you borrow many techniques from neural contextual linear bandit [Zhou et al. 2020] and NTK [Arora et al. (2019b); Cao & Gu (2019b); Allen-Zhu et al. (2019b)] to execute the high-level idea of decoupling representation learning and exploration from [Riquelme et al. (2018)] in theoretical analysis. However, I think that the contribution and novelty is weak”
> > > **A**:  The NTK analysis in [Arora et al. (2019b); Cao & Gu (2019b); Allen-Zhu et al. (2019b)] has become one of the most common techniques to analyze the approximation error and convergence properties of over-parameterized neural networks. It has been successfully applied to different problems such as reinforcement learning [1, 2], contextual bandits [3, Zhou et al. 2020], online learning [4], etc. Therefore, we do not think the usage of NTK in our specific problem setting diminishes the contribution and novelty of our work.
> > >
> > > Compared with [Zhou et al. 2020], we do not use the whole parameter in the reward estimator to construct the UCB bonus. Note that using the whole parameter in UCB is more straightforward since it mimics the way how linear UCB constructs the exploration bonus. In contrast, it is more challenging to derive the regret bound when the parameter in the reward estimator is decomposed into two parts since we need to show that with partial information we can still prove
> > > (1) the network parameter up to the last hidden layer converges, and (2) the last layer parameter is in a confidence ball of the true parameter. No existing theoretical analysis can deal with the setting in our paper since all of them rely on the full exploration of all weight parameters. Therefore, our analysis is very different from any existing work in neural contextual bandits.
> > >
> > > Last but not least, we think borrowing some proof techniques from prior work is a good thing rather than a bad thing. It is a common practice for a theoretical paper to prove that a new algorithm/idea works by combining existing proof techniques and new proof techniques. This is how research makes progress.
> > >
> > > References:
> > > [1] Qi Cai, Zhuoran Yang, Jason D Lee, and Zhaoran Wang. Neural temporal-difference learning converges to global optima. In Advances in Neural Information Processing Systems, 2019.
> > > [2] Lingxiao Wang, Qi Cai, Zhuoran Yang, and Zhaoran Wang. Neural policy gradient methods: Global optimality and rates of convergence. In International Conference on Learning Representations, 2020.
> > > [3] Zhang, Weitong, Dongruo Zhou, Lihong Li, and Quanquan Gu. "Neural Thompson Sampling." In International Conference on Learning Representations. 2021.
> > > [4] Chen, X., Minasyan, E., Lee, J.D. and Hazan, E., 2021. Provable Regret Bounds for Deep Online Learning and Control. arXiv preprint arXiv:2110.07807.

---

### Official Review · Reviewer_pfBo · 2021-11-04

**Correctness:** 4
**Technical Novelty And Significance:** 3
**Empirical Novelty And Significance:** 2
**Recommendation:** 8
**Confidence:** 3

**Main Review:**


Strengths:
- The paper is well written, algorithmic motivations are clear, and the setting is interesting.
- The theoretical results, which are the main contribution, are nice and the assumptions made are very clearly described and contextualized.
- Experimental results consider several reasonable domains+baselines and support the claims.

Weaknesses:
- The main weakness is that the algorithm proposed is not particularly novel, since (Riquelme et al. 2018) already introduce the "decoupled deep representation+last layer search" methodology in a previous paper. So the question is whether the new regret analysis is enough on its own. For instance, the results are given for UCB only -- could a similar analysis be extended to the Thompson sampling case (i.e., that used in Riquelme et al. 2018)?
- There is also some concern about the requirement on the width of the neural network in thm 4.4 being so large in order for the main O(\sqrt(T)) bound to hold, rather than a maybe more honest O(m^{-1/6}T) bound, but the authors do at least provide some empirical evidence in the appendix that suggests the performance of their algorithm is not heavily reliant on very large values of the network width.

-------------------------------
Update: I have read the author response. They adequately address my few minor concerns and so my opinion to accept stays the same.

**Summary Of The Paper:**

Authors tackle the setting of contextual bandits, using deep representation learning combined with an upper confidence bound algorithm. The main contribution of this work is to provide a regret bound for the setup which decouples the representation learning from the UCB search, by searching only over the last layer of the network. The setting had been studied before, but only empirically, and using Thompson sampling rather than UCB. Authors validate their results empirically on several domains from the UCI data repo, as well as on MNIST, comparing against state of the art baselines.


**Summary Of The Review:**

I am recommending accept because the theoretical results are interesting, and combined with the given very clear discussion about the assumptions in context, I think provide a decent baseline for future such analyses, even if assumptions such as the large network width are restrictive.

---

> ### Author Response · Authors · 2021-11-18
> **Response to Reviewer pfBo**
>
> Thank you for your helpful comments. We address your comments point by point as follows.
>
> ---
> **Q**: “The results are given for UCB only -- could a similar analysis be extended to the Thompson sampling case (i.e., that used in Riquelme et al. 2018)?”
> **A**: Thank you for this great question. The proof technique in our paper is the first one to deal with neural bandit algorithms with decoupled deep representation learning and shallow exploration. It is not limited to UCB-type of exploration. We believe it could be extended to the Thompson sampling type of algorithms with nontrivial modification. In particular, we need to change the algorithm and will have to modify the proof framework in Theorem 4.4 by following the argument of saturated and unsaturated arms used in Agrawal and  Goyal (2013) (Thompson sampling for contextual bandits with linear payoffs." ICML).
>
> ---
> **Q**: “There is also some concern about the requirement on the width of the neural network in thm 4.4 being so large in order for the main O(\sqrt(T)) bound to hold, rather than a maybe more honest O(m^{-1/6}T) bound, but the authors do at least provide some empirical evidence in the appendix that suggests the performance of their algorithm is not heavily reliant on very large values of the network width.”
> **A**: Thank you for your supportive comment on our empirical study on the width of the neural network used in our theory.  Indeed, in practice we don’t need very wide neural networks to achieve small regrets. The strong requirement on the neural network width is due to the limitation of the NTK-based analysis for deep learning. Note that the main contribution of our work is that it is the first to show that the idea of deep representation learning and shallow exploration works and achieves a sublinear regret.

---

### Official Review · Reviewer_16Sn · 2021-11-07

**Correctness:** 4
**Technical Novelty And Significance:** 3
**Empirical Novelty And Significance:** 2
**Recommendation:** 6
**Confidence:** 4

**Main Review:**

Strengths:
- The authors combine existing techniques to create an interesting algorithm for contextual bandits. Although the combination is not novel, the authors make a theoretical contribution by demonstrating that the combination has sub-linear O(\sqrt(T)) regret.
- The paper is well written, limitations are clearly stated.
- The authors show that their proposed algorithm outperforms the existing algorithms.

Weaknesses:
- Is the direct comparison with the existing neural bandit algorithms possible? It appears that this paper has an additional assumption than the existing literature on neural bandits. I am not sure even the regret bounds can be compared.
- The gap between the theoretically suggested setting and the experiment: there seems to be a huge gap between what is shown theoretically and what is executed in the experiments, for example, $m$. Can you show at least some experiments on how the algorithm behaves with a very large value of  $m$ which is what the theoretical results are based on?


**Summary Of The Paper:**

The paper presents a new neural-bandit algorithm with shallow exploration and provides a regret bound for the proposed method. The existing approaches have introduced deep neural networks based bandit algorithms to learn reward functions, in which exploration takes place over the entire network parameter space, which can be inefficient for large-size networks which are typical in NTK based approaches. The authors address this by taking an existing approach that decouples the deep neural network feature representation learning from most of the exploration of the network parameters by only exploring over the final layer of the network.

Despite the fact that this idea of shallow exploration has been proposed previously, there has not been a theoretical analysis with a regret bound. The authors analyze a UCB version of this approach, then build from techniques from both deep neural contextual bandits and linear contextual bandits to prove an O(\sqrt(T)) regret bound. Finally, the authors present experimental results to show that their algorithm work well in practice.

**Summary Of The Review:**

With the points I made above, I am leaning slightly on acceptance. I hope the authors address the concerns mentioned above.

========== Post-reponses =================

Thanks to the authors for the responses. I have read the comments of the other reviewers. I am staying with the current score.

---

> ### Author Response · Authors · 2021-11-18
> **Response to Reviewer 16Sn**
>
> Thank you for your helpful comments. We address your comments point by point as follows.
>
> ---
> **Q**: “Is the direct comparison with the existing neural bandit algorithms possible? It appears that this paper has an additional assumption than the existing literature on neural bandits. I am not sure even the regret bounds can be compared.”
> **A**: Indeed, the regret bound of our algorithm and that of existing algorithms are not directly comparable, because our result needs an additional assumption, i.e., Assumption 4.2.  We would like to emphasize that the goal of this paper is to show that decoupling representation learning and exploration can still achieve sublinear regret, by proposing a new algorithm and a new regret analysis. The main advantage of decoupling representation learning and exploration is its computational efficiency. We did not try to argue that the regret of our algorithm is better than those of other neural bandit algorithms.
>
> ---
> **Q**: “The gap between the theoretically suggested setting and the experiment: there seems to be a huge gap between what is shown theoretically and what is executed in the experiments, for example, $m$. Can you show at least some experiments on how the algorithm behaves with a very large value of $m$  which is what the theoretical results are based on?”
> **A**: Thank you for your suggestion. We indeed conducted experiments with large values of $m$ to see how our algorithm performs. The results were presented in Appendix A.2, where we show that our algorithm works better when the value of $m$ increases. This is consistent with our theoretical results. Currently, we are also expanding this experiment with even larger values of $m$. We will update the reviewer with our new results as soon as we have them.
>
> On the other hand, we would also like to point out that the theoretical value of $m$ is a sufficient condition to prove our results. However, we believe it is not necessary in practice. In other words, our theory may be conservative since our current analysis is based on the current limited understanding of deep learning. Nevertheless, we believe our work is a good and unique addition to the current neural network-based bandit literature.

---

> ### Author Response · Authors · 2021-11-22
> **New experiments are added to the revision**
>
> Thank you for your suggestion of adding new experiments with larger values of m (the width). We have added a new experiment in Appendix A.2, where the width of the network is set to 50,000. The results are presented in Figure 2. As is anticipated from our theory, when $m$ increases, the performance of our algorithm improves. And when $m$ is very large, the regret will be dominated by $\tilde{O}(\sqrt{T})$.

---

### Decision · Program_Chairs · 2022-01-20

**Decision:**

Accept (Poster)

**Comment:**

This paper tackles the neural contextual bandit problem, for which existing approaches consists rely on bandit algorithms based on deep neural networks to learn reward functions. In these existing strategies, exploration takes place over the entire network parameter space, which can be inefficient for the large-size networks typically used in NTK-based approaches. In this work, the authors address this by building on an existing technique of shallow exploration, which consists in exploring over the final layer of the network only, allowing to decouple the deep neural network feature representation learning from most of the exploration of the network parameters. More specifically, they propose a simple and effective UCB-based strategy using this shallow exploration scheme, for which they provide a theoretical analysis. The proposed approach builds on several ideas for previous works, including borrowing proof techniques and theoretical arguments. Although this limits the novelty of the work, connecting these ideas together is not obvious and constitutes a significant contribution. Moreover, the proposed approach fixes an important known issue due to the matrix inversion in LinUCB, which could have a strong impact on the bandit community.